# Multi-Task Learning with Summary Statistics

**Parker Knight**
Department of Biostatistics
Harvard University
Boston, MA
pknight@g.harvard.edu

**Rui Duan**
Department of Biostatistics
Harvard University
Boston, MA
rduan@hsph.harvard.edu

## Abstract

Multi-task learning has emerged as a powerful machine learning paradigm for integrating data from multiple sources, leveraging similarities between tasks to improve overall model performance. However, the application of multi-task learning to real-world settings is hindered by data-sharing constraints, especially in healthcare settings. To address this challenge, we propose a flexible multi-task learning framework utilizing summary statistics from various sources. Additionally, we present an adaptive parameter selection approach based on a variant of Lepski's method, allowing for data-driven tuning parameter selection when only summary statistics are available. Our systematic non-asymptotic analysis characterizes the performance of the proposed methods under various regimes of the sample complexity and overlap. We demonstrate our theoretical findings and the performance of the method through extensive simulations. This work offers a more flexible tool for training related models across various domains, with practical implications in genetic risk prediction and many other fields.

## 1  Introduction

The growing availability of extensive and intricate datasets presents an opportunity to integrate data from multiple sources. Multi-task learning has emerged as a promising machine learning approach that enables the simultaneous learning of multiple related models, leveraging shared structure between tasks to enhance the performance on each task individually [36, 35]. In healthcare and biomedical research, the practical application of multi-task learning is often hindered by data-sharing constraints, which stem from concerns about the ownership and privacy of individual-level data [32, 8]. Patient data in these domains is typically sensitive and less likely to be publicly available or shared across study sites, limiting researchers' access to individual-level data from different domains.

To overcome this limitation, researchers have increasingly integrated summary statistics into analysis pipelines as a substitute for individual-level data [22, 13, 12]. Summary statistics are straightforward, interpretable measures derived from raw data that can offer insights into data distribution, variability, and relationships among variables. Furthermore, they can be aggregated across studies to facilitate data integration and reused in various research projects. Recently, the use of summary statistics has garnered interest in healthcare and biomedical research. For example, many genetic risk prediction methods rely on summary-level statistics such as associations from Genome-wide Association Studies (GWAS), Linkage Disequilibrium estimations (LD), and minor allele frequencies (MAFs) [6]. These summary statistics can help predict an individual's likelihood of developing specific diseases based on their genetic profile.

Inspired by a potential use case in genetic risk prediction, we propose a multi-task learning framework that enables simultaneous learning of multiple genetic risk prediction models using only publicly available summary statistics. Our proposed framework can be used in the context of predicting genetic risks for multiple traits leveraging potentially shared genetic pathways, and can also be used

to develop trans-ethnic genetic risk prediction models that account for potential heterogeneity across populations, improving generalizability and real-world applicability. Beyond genetic risk prediction, the ability to learn from summary statistics offers a versatile tool for developing models across a wide range of domains, including healthcare, finance, and marketing.

To summarize, the contributions of this work are threefold: First, we propose a flexible multi-task learning framework which allows training multiple models simultaneously using basic summary statistics characterizing marginal relationship between outcomes and features, which are often publicly available. We allow summary statistics corresponding to each task to be generated from distinct or potentially overlapping samples. Secondly, we conducted a systematic non-asymptotic analysis which characterizes how the performance of the proposed methods are influenced by the characteristics of summary statistics. In particular, we show that there are multiple regimes of performance depending on the sample complexity of the source datasets and their overlap. The theoretical results are supported with extensive simulations. Lastly, We propose an adaptive scheme for tuning parameter selection based on the variant of Lepski's method [14] given in [5]. This allows us to select a data-driven tuning parameter when only summary statistics are available and cross-validation is not feasible. We prove that tuning parameters chosen by this method satisfy an oracle inequality with high probability, and demonstrate the effectiveness of the method via simulations.

## 1.1 Related work

The use of summary statistics for regression modeling has been considered in the statistical genetics literature [6]. The `lassosum` method for polygenic risk prediction was introduced by [22], which considered fitting a $L_1$ penalized linear regression with summary statistics, and its theoretical properties were studied in depth by [15]. In [4], the authors extend these ideas to polygenic risk prediction with binary traits. The summary statistics used in these methods include the marginal associations between genetic variants and phenotypes, and statistics summarizing the covariance structures among all genetic variants oftentimes derived from a reference genotype dataset. Empirical studies have demonstrated that the efficacy of such models is significantly influenced by the choices of the GWAS summary statistics and the reference dataset [30]. However, there's still limited theoretical understanding regarding how the overlap of samples and the inherent heterogeneity between datasets impact the model performance. Moreover, most current approaches devise models for a single trait within a single ancestral population. Considering shared genetic architectures could potentially enhance performance by employing a multi-task learning strategy [23].The authors of [24] take this approach, and describe a multi-task estimator for multi-ancestry pQTL analysis. However, they do not consider the setting when only summary statistics are available for each task.

Our methods build upon classical multi-task learning techniques, and enable fitting models only using basic summary statistics which are often made publicly available. The sparse regularized estimator extends the group-sparse estimators studied in [19, 18], while the nuclear norm estimator expands on the low-rank regression model described in [25]. The nuclear norm approach is closely related to the linear representation learning problem [7, 31], which constrains the regression coefficients to a shared low-dimensional subspace.

Another recent line of work studies the multi-task learning problem under data-sharing constraints. In [17], the authors describe a federated multi-task learning linear regression model for privacy-preserving data analysis. Similarly, [2] presents a computational framework for multi-task learning under DataSHIELD [11] constraints. The formulation of these methods is conceptually similar to ours, but they do not provide theoretical guarantees for their estimators, and we consider a more flexible setting where the summary statistics can be derived from different sources.

Finally, our methods are closely related to the one-shot federated learning paradigm, in which only one round of communication is permitted between the primary local research site and additional sites. [9] presents an algorithm for fitting logistic regression models using summary statistics from different research sites. The works of [20, 34] extend these ideas to linear mixed effects models and generalized mixed effects models, respectively. [21] presents a federated algorithm for fitting the Cox proportional hazards model, and [16] studies federated transfer learning methods for fitting generalized linear models. Nevertheless, the summary statistics addressed in our research are frequently reported in existing studies and can be employed across various models. This is in contrast to the one-shot federated algorithm, where the summary statistics are model specific and the implementation relies on the infrastructure of a collaborative environment.

## 2    Problem setup and methods

Consider the setting where we are interested in learning a total of $Q$ tasks simultaneously. For each $q \in [Q]$, we posit the linear model

$$\mathbf{Y}^{(q)} = \mathbf{X}^{(q)}\beta^{(q)} + \varepsilon^{(q)}$$

where $\mathbf{Y}^{(q)} \in \mathbb{R}^{n_q}$, $\mathbf{X}^{(q)} \in \mathbb{R}^{n_q \times p}$, and $\varepsilon^{(q)}$ is mean-zero random noise. Each index $q$ corresponds to the $q_{th}$ task. The dataset $\mathcal{D}^{(q)} = (\mathbf{Y}^{(q)}, \mathbf{X}^{(q)})$ contains the individual-level observations of the outcome and features respectively for the $q_{th}$ task. We consider the generic setting where the features $\mathcal{D}^{(q)}$ might be collected from either overlapping or non-overlapping samples across tasks. Our estimand of interest is the matrix $\mathbf{B}^* = [\beta^{(1)}, ..., \beta^{(Q)}] \in \mathbb{R}^{p \times Q}$, where the $q_{th}$ column of $\mathbf{B}^*$ is $\beta^{(q)}$. Furthermore, let $e_i$ denote the $i_{th}$ standard basis vector, so that $\beta^{(q)} = \mathbf{B}^* e_q$.

If all the individual-level observations $\mathcal{D}^{(q)}$ are available, a natural estimator of $\mathbf{B}^*$ is the regularized multi-task least-squares estimator

$$\widehat{\mathbf{B}} = \arg\min_{\mathbf{B}} \left\{ \sum_{q \in [Q]} \frac{1}{2n_q} \left\| \mathbf{Y}^{(q)} - \mathbf{X}^{(q)}\mathbf{B}e_q \right\|_2^2 + \lambda\mathcal{P}(\mathbf{B}) \right\} \tag{1}$$

where $\mathcal{P}$ is a suitable penalty, chosen to enforce similarity structure between tasks, with tuning parameter $\lambda > 0$. However, in many applications, we are less likely to observe $\mathcal{D}^{(q)}$. Rather, summary statistics which contains information of the feature-outcome and feature-feature relationships may be more likely to be made publicly available. Motivated by the use case in genetic risk prediction, we assume that only summary statistics $\widehat{\mathbf{S}}^{(q)}$ and $\widetilde{\mathbf{\Sigma}}^{(q)}$ are observable, where $\widehat{\mathbf{S}}^{(q)} = \frac{1}{n_q}\mathbf{X}^{(q)T}\mathbf{Y}^{(q)}$ are derived from $\{\mathbf{X}^{(q)}, \mathbf{Y}^{(q)}\}$, which we termed as the discovery data, and $\widetilde{\mathbf{\Sigma}}^{(q)} = \frac{1}{\tilde{n}_q}\widetilde{\mathbf{X}}^{(q)T}\widetilde{\mathbf{X}}^{(q)}$ is a sample covariance matrix computed from the proxy data $\widetilde{\mathbf{X}}^{(q)} \in \mathbb{R}^{\tilde{n}_q \times p}$, which may or may not have overlap with $\mathbf{X}^{(q)}$.

Our goal is to estimate $\mathbf{B}^*$ using two sets of summary statistics $\widehat{\mathbf{S}}^{(q)}$ and $\widetilde{\mathbf{\Sigma}}^{(q)}$. We note that $\widetilde{\mathbf{X}}^{(q)}$ is not necessarily equal to $\mathbf{X}^{(q)}$. In practice, the studies which report $\widehat{\mathbf{S}}^{(q)}$ may not be the same as the ones reporting $\widetilde{\mathbf{\Sigma}}^{(q)}$. Intuitively, we hope that $\widetilde{\mathbf{X}}^{(q)}$ is generated from a similar population as $\mathbf{X}^{(q)}$, but this may not hold in general. In Section 3, our theoretical analysis reveals how the overlap between $\widetilde{\mathbf{X}}^{(q)}$ and $\mathbf{X}^{(q)}$ and their distributional shift can influence the accuracy of multi-task learning.

To construct an estimator that uses only the information provided by $\widetilde{\mathcal{D}}^{(q)}$, we notice that the least-squares loss can be written as

$$\mathcal{L}(\beta) = \|\mathbf{Y} - \mathbf{X}\beta\|_2^2 = \mathbf{Y}^T\mathbf{Y} - 2\langle\beta, \mathbf{X}^T\mathbf{Y}\rangle + \beta^T\mathbf{X}^T\mathbf{X}\beta$$

By dropping the constant term, we arrive at a loss function that can be computed using only summary-level information, namely the matrices $\mathbf{X}^T\mathbf{Y}$ and $\mathbf{X}^T\mathbf{X}$. This motivates our general strategy for constructing an estimator only using summary statistics: we substitute $\widehat{\mathbf{S}}^{(q)}$ and $\widetilde{\mathbf{\Sigma}}^{(q)}$ where appropriate in each least-square loss function in Equation 1 and arrive at the following optimization problem.

$$\widehat{\mathbf{B}} = \arg\min_{\mathbf{B}} \left\{ \sum_{q \in [Q]} \frac{1}{2} \left\| \widetilde{\mathbf{\Sigma}}^{(q)1/2}\mathbf{B}e_q \right\|_2^2 - \langle\widehat{\mathbf{S}}^{(q)}, \mathbf{B}e_q\rangle + \lambda\mathcal{P}(\mathbf{B}) \right\} \tag{2}$$

There are many possible choices of $\mathcal{P}$ for enforcing structure similarities across tasks. For instance, the recent works of [29] and [12] study low-rank and angle-based penalties for enforcing a shared orientation among the task-specific parameters. In this work, we study two estimators obtained under the $\ell_{2,1}$ norm penalty, denoted $\|.\|_{2,1}$, and the nuclear norm penalty, denoted $\|.\|_*$. These penalties are chosen for their intuitive interpretation: the $\ell_{2,1}$ penalty is more likely to be effective if a common set of variables are active across the tasks. If the task-specific parameters tend to be "correlated",

in the sense that they lie in a low-dimensional subspace, the nuclear norm penalty is preferred. In practice, certain domain knowledge can be incorporated to determine the penalty structure, or it can be chosen in a data-driven way in the existence of a validation dataset.

The corresponding estimators are expressed as follows:

$$\widehat{\mathbf{B}}^{(sp)} = \arg\min_{\mathbf{B}} \left\{ \sum_{q \in [Q]} \frac{1}{2} \left\| \widetilde{\mathbf{\Sigma}}^{(q)1/2} \mathbf{B} e_q \right\|_2^2 - \langle \widehat{\mathbf{S}}^{(q)}, \mathbf{B} e_q \rangle + \lambda \left\| \mathbf{B} \right\|_{2,1} \right\} \tag{3}$$

$$\widehat{\mathbf{B}}^{(lr)} = \arg\min_{\mathbf{B}} \left\{ \sum_{q \in [Q]} \frac{1}{2} \left\| \widetilde{\mathbf{\Sigma}}^{(q)1/2} \mathbf{B} e_q \right\|_2^2 - \langle \widehat{\mathbf{S}}^{(q)}, \mathbf{B} e_q \rangle + \lambda \left\| \mathbf{B} \right\|_* \right\} \tag{4}$$

The superscripts $(sp)$ and $(lr)$ stand for "sparse" and "low-rank" respectively.

## 3   Theoretical guarantees

Before presenting our theoretical results, we first introduce the relevant notation. Let $N$ denote the total size of discovery observations and proxy observation across all $Q$ tasks. Formally,

$$N = \sum_{q=1}^{Q} (n_q + \tilde{n}_q)$$

We note that $N$ may double-count individuals who are part of both the proxy data and the discovery data. Define the subset $\mathcal{I}_q \subset [N]$ as the index set for the discovery data points in the $q_{th}$ task; in other words $i \in \mathcal{I}_q$ implies $X_i \in \mathbb{R}^p$ is a row of $\mathbf{X}^{(q)}$. We define $\widetilde{\mathcal{I}}_q$ analogously for the proxy data; $i \in \widetilde{\mathcal{I}}_q$ implies $X_i$ is a row of $\widetilde{\mathbf{X}}^{(q)}$. Let $\tilde{\rho}_q = |\mathcal{I}_q \cap \widetilde{\mathcal{I}}_q|/\tilde{n}_q$ denote the proportion of proxy samples which are also in the discovery dataset for the $q_{th}$ task. In the results that follow, let

$$\gamma_q = 1 + \left\| \beta^{(q)} \right\|_2^2 \left( \frac{n_q}{\tilde{n}_q} + 1 - 2\tilde{\rho}_q \right)$$

and take $\gamma = \max_q \gamma_q$. Additionally, let $\Xi \in \mathbb{R}^{p \times Q}$ be the matrix with its $q_{th}$ column equal to $(\mathbf{\Sigma}_1^{(q)} - \mathbf{\Sigma}_2^{(q)})\beta^{(q)}$, where $\mathbf{\Sigma}_1^{(q)}$ and $\mathbf{\Sigma}_2^{(q)}$ are the population-level covariance matrices of $\mathbf{X}^{(q)}$ and $\widetilde{\mathbf{X}}^{(q)}$ respectively. The quantities $\gamma$ and $\Xi$ play important roles in our results that follow. In particular, $\gamma$ is a multiplicative factor in our bounds that represents the cost of using proxy data rather than individual-level data. Similarly, $\Xi$ will represent the cost of using a proxy dataset with a distributional shift from the discovery data. Finally, we will let $n_{\min}$ and $\tilde{n}_{\min}$ denote the smallest sample size of discovery and proxy data, respectively. All proofs are given in the supplement.

### 3.1   Guarantees for $\ell_{2,1}$-norm estimator

In this section, we formally state our assumptions and results for the $\widehat{\mathbf{B}}^{(sp)}$ estimator. The assumptions are standard for high-dimensional regularized estimators, see [26] for a deeper discussion of these conditions.

**Assumption 3.1** (Sub-gaussian design and noise). *The following holds for each $q \in [Q]$: The rows of $\mathbf{X}^{(q)}$ are independent and identically distributed according to a sub-Gaussian distribution with covariance matrix $\mathbf{\Sigma}_1^{(q)} \in \mathbb{R}^{p \times p}$. Similarly, the rows of $\widetilde{\mathbf{X}}^{(q)}$ are independent and identically distributed according to a sub-Gaussian distribution with covariance $\mathbf{\Sigma}_2^{(q)} \in \mathbb{R}^{p \times p}$. The matrices $\mathbf{\Sigma}_1^{(q)}$ and $\mathbf{\Sigma}_2^{(q)}$ have bounded eigenvalues. The entries of $\varepsilon^{(q)}$ are independent and identically distributed according to a sub-Gaussian distribution with parameter $\sigma^2$. The $\mathbf{X}^{(q)}$ and $\varepsilon^{(q)}$ are independent of one another.*

**Assumption 3.2** (Shared support). *There exists a subset $S^* \subset [p]$ such that $\mathsf{supp}(\beta^{(q)}) = S^*$ for each $q$.*

**Definition 3.1** (Sparse cone). *For any $S \subset [p]$, let*

$$\mathcal{C}_\alpha(S) = \left\{ \Delta \in \mathbb{R}^{p \times Q} : \|\Delta_{S^c}\|_{2,1} \leq \alpha \|\Delta_S\|_{2,1} \right\}$$

**Assumption 3.3** (Restricted strong convexity). *There exists a constant $\kappa > 0$ and a sequence $a_N \to 0$ as $N \to \infty$ such that the following inequality holds for each $\Delta \in \mathcal{C}_3(S^*)$ with probability at least $1 - a_N$:*

$$\sum_{q=1}^{Q} \left\| \widetilde{\boldsymbol{\Sigma}}^{(q)1/2} \Delta e_q \right\|_2^2 \geq \frac{1}{\kappa} \|\Delta\|_F^2$$

**Theorem 3.1.** *Under assumptions 3.1, 3.2, and 3.3, there exist constants $c_1$ and $c_2$ depending only on the $\sigma^2$ and the eigenvalues of $\boldsymbol{\Sigma}_1^{(q)}$ and $\boldsymbol{\Sigma}_2^{(q)}$ such that if $n_{\min} \wedge \tilde{n}_{\min} \geq c_1 \|\mathbf{B}^*\|_{\infty,\infty} (Q + \log p)$ and $\lambda = O(\sqrt{\gamma(Q + \log p)/n_{\min}} + \|\Xi\|_{2,\infty})$, the following inequality holds with probability at least $1 - e^{-\log p} - a_N$:*

$$\left\| \widehat{\mathbf{B}}^{(sp)} - \mathbf{B}^* \right\|_F \leq c_2 \left( \sqrt{\frac{\gamma s(Q + \log p)}{n_{\min}}} + \sqrt{s} \|\Xi\|_{2,\infty} \right)$$

In the subsequent discussion, we take $q^* = \arg\max_{q \in [Q]} \gamma_q$ and $(n, n, \tilde{\rho}) = (n_{q^*}, \tilde{n}_{q^*}, \tilde{\rho}_{q^*})$ so that the triplet $(n, \tilde{n}, \tilde{\rho})$ corresponds to the same sizes and overlap factor used to compute $\gamma$.

There are three main quantities in this upper bound that are of novel interest: the ratio of discovery data size to proxy data size $n/\tilde{n}$, the proportion of overlap between the discovery and proxy data $\tilde{\rho}$, and the error in specifying the proxy data distribution $\Xi = (\boldsymbol{\Sigma}^{(1)} - \boldsymbol{\Sigma}^{(2)})\beta$. The first two of these are captured by the factor $\gamma$. Our results show that with fixed $n$ and $\tilde{n}$, the larger proportion of overlap leads to better estimation accuracy. When the proxy data and discovery data are precisely the same, meaning that $\mathcal{I}_q = \tilde{\mathcal{I}}_q$ for all $q$, we recover the minimax rate of estimation for the $\ell_{2,1}$ penalized multi-task learning problem established by Theorem 6.1 of [18]. If the proxy data and the discovery data are disjoint, meaning that $\mathcal{I}_q \cap \tilde{\mathcal{I}}_q = \emptyset$ for all $q$, the error is increased relative to the minimax rate by a factor of $(1 + n/\tilde{n}) \|\beta\|_2^2$. This recovers the result of Theorem 2.1 in [15] up to a constant factor, assuming that $\Xi = 0$. The novelty of Theorem 3.1 is that we are able to characterize the convergence rate of $\widehat{\mathbf{B}}^{(sp)}$ for any values of $n/\tilde{n}$, $\tilde{\rho}$, and $\Xi$. Additionally, we emphasize that the form of the $\gamma$ term implies that a price is paid anytime when $\mathbf{X}^{(q)}$ is not fully contained in $\widetilde{\mathbf{X}}^{(q)}$. Indeed, if $\tilde{\rho} < 1/2$ and we take $\tilde{n} \to \infty$ we still have that $\gamma > 1$ as long as the signal is nonzero. Counter-intuitively, this indicates that an oracle model which has full access to the population-level covariance matrix of the covariates will perform worse in terms of estimation error than an estimator which has access to individual-level data. Furthermore, if $\tilde{\rho} > 1/2$, our theorem predicts that the estimator will out-perform the oracle estimator that uses the population covariance matrix. These phenomena are validated in our simulation studies in Section 5.

## 3.2 Guarantees for the nuclear norm estimator

Now we state our results for the low-rank proxy data estimator, when the penalty is taken to be the nuclear norm. Once again, these assumptions are standard for high-dimensional regression problems with the nuclear norm [26, 33].

**Assumption 3.4** (Low rank). *The matrix $\mathbf{B}^*$ has rank $r << p \wedge Q$. Let $\mathcal{U}^*$ and $\mathcal{V}^*$ denote the column space and row space of $\mathbf{B}^*$ respectively. Note that $\mathcal{U}^*$ and $\mathcal{V}^*$ each have dimension $r$.*

**Definition 3.2** (Subspaces). *Let $\mathcal{U}$ denote a dimension $k \leq p \wedge Q$ subspace of $\mathbb{R}^p$, and let $\mathcal{V}$ denote a dimension $k \leq p \wedge Q$ subspace of $\mathbb{R}^Q$. Define*

$$\mathbb{M} = \mathbb{M}(\mathcal{U}, \mathcal{V}) := \left\{ \Delta \in \mathbb{R}^{p \times Q} : \mathsf{row}(\Delta) = \mathcal{V}, \mathsf{col}(\Delta) = \mathcal{U} \right\}$$

$$\mathbb{M}^{\perp} = \mathbb{M}^{\perp}(\mathcal{U}, \mathcal{V}) = \left\{ \Delta \in \mathbb{R}^{p \times Q} : \mathsf{row}(\Delta) \perp \mathcal{V}, \mathsf{col}(\Delta) \perp \mathcal{U} \right\}$$

*Furthermore, for any subspace $\Omega$ of $\mathbb{R}^{p \times Q}$, let $\Delta_{\Omega}$ denote the projection of $\Delta$ onto $\Omega$.*

*We will denote $\mathbb{M}^* = \mathbb{M}(\mathcal{U}^*, \mathcal{V}^*)$.*

**Definition 3.3** (Low rank cone). *For any set $\mathbb{M}$ as defined above, let*

$$\mathcal{C}_{\alpha}(\mathbb{M}) = \left\{ \Delta \in \mathbb{R}^{p \times Q} : \|\Delta_{\mathbb{M}^{\perp}}\|_* \leq \alpha \|\Delta_{\mathbb{M}}\|_* \right\}$$

**Assumption 3.5** (Restricted strong convexity). *There exists a constant $\kappa > 0$ and a sequence $b_N \to 0$ as $N \to \infty$ such that the following inequality holds for each $\Delta \in \mathcal{C}_3(\mathbb{M}^*)$ with probability at least $1 - b_N$:*

$$\sum_{q=1}^{Q} \left\| \widetilde{\boldsymbol{\Sigma}}^{(q)1/2} \Delta e_q \right\|_2^2 \geq \frac{1}{\kappa} \|\Delta\|_F^2$$

**Theorem 3.2.** *Under assumptions 3.1, 3.4, and 3.5, there exist constants $c_1$ and $c_2$ depending only on $\sigma^2$ and the eigenvalues of $\boldsymbol{\Sigma}_1^{(q)}$ and $\boldsymbol{\Sigma}_2^{(q)}$ such that if $n_{\min} \wedge \tilde{n}_{\min} \geq c_1 \|\mathbf{B}^*\|_{\infty,\infty} (Q + p)$ and $\lambda = O(\sqrt{\gamma(Q+p)/n_{\min}} + \|\Xi\|_{\mathsf{op}})$, the following inequality holds with probability at least $1 - e^{-p} - b_N$:*

$$\left\| \widehat{\mathbf{B}}^{(lr)} - \mathbf{B}^* \right\|_F \leq c_2 \left( \sqrt{\frac{r\gamma(Q+p)}{n_{\min}}} + \sqrt{r} \|\Xi\|_{\mathsf{op}} \right)$$

This theorem recovers precisely the same behavior with respect to $\gamma$ and $\Xi$ as Theorem 3.1. As $\gamma \to 1$, we achieve the minimax rate of estimation for low-rank regression as derived in [27] as long as $\Xi = 0$.

## 4 Tuning parameter selection with Lepski's method

A key challenge of applying penalized regression models to summary statistics is that model tuning based on data splitting (e.g., training and validation) is no longer an option. Model selection methods based on information criteria require knowing the log squared loss $\log \|\mathbf{Y}^{(q)} - \mathbf{X}^{(q)}\beta\|_2^2$, which cannot be recovered from $\widehat{\mathbf{S}}^{(q)}$ and $\widetilde{\boldsymbol{\Sigma}}^{(q)}$ [1]. To address this, we propose to use a tuning scheme based on Lepski's method [14], a classical tool of nonparametric statistics for adaptive estimation with unknown tuning parameters. The authors of [5] apply the ideas of Lepski to the LASSO, providing a fast algorithm for model tuning with non-asymptotic guarantees. In this section, we extend the methods in [5] to tune the multi-task estimators described in the present work.

The results and ideas in this section apply to both $\widehat{\mathbf{B}}^{(sp)}$ and $\widehat{\mathbf{B}}^{(lr)}$, so without loss of generality, let $(\widehat{\mathbf{B}}_{\lambda}, \mathcal{P})$ denote a generic estimator-regularizer pair with tuning paramter $\lambda$, which may refer to either $(\widehat{\mathbf{B}}_{\lambda}^{(sp)}, \|.\|_{2,1})$ or $(\widehat{\mathbf{B}}_{\lambda}^{(lr)}, \|.\|_*)$. Additionally, let $\mathcal{P}^*$ denote the dual of $\mathcal{P}$, meaning that

$$\mathcal{P}^*(X) = \sup_{Y:\mathcal{P}(Y) \leq 1} \langle X, Y \rangle$$

Finally, we let $\mathcal{L}$ denote the loss function for both estimators and let $\nabla \mathcal{L}$ denote its gradient.

The intuition behind the adaptive tuning procedure is that the tuning parameter should be chosen large enough to control fluctuations in the gradient of the loss function, but not too large such that too much bias is incurred. [26] articulates that the performance of regression estimator with a convex penalty is contingent on the following event occurring with high probability:

$$\mathcal{A}(\lambda) = \left\{ \mathcal{P}^*(\nabla \mathcal{L}(\mathbf{B}^*)) \leq \frac{\lambda}{2} \right\}$$

where we use our problem's notation for continuity. The proofs of Theorem 3.1 and 3.2 involve showing that $\mathcal{A}(\lambda)$ holds with high probability under our stated conditions. It is straightforward to prove the following proposition, which states that conditional on $\mathcal{A}$, the gradient of the loss function at our generic estimator $\widehat{\mathbf{B}}$ is close to the gradient at the true parameter $\mathbf{B}^*$.

**Proposition 4.1.** *Let $(\widehat{\mathbf{B}}_\lambda, \mathcal{P})$ denote a generic estimator-regularizer pair. Conditional on the event $\mathcal{A}(\lambda)$, there exists a constant $C > 0$ such that the following inequality is satisfied almost surely:*

$$\mathcal{P}^*(\nabla \mathcal{L}(\widehat{\mathbf{B}}_\lambda) - \nabla \mathcal{L}(\mathbf{B}^*)) \leq C\lambda$$

This proposition motivates the following definition, which we adopt from [5].

**Definition 4.1.** *Let $\Lambda = \{\lambda_1, \lambda_2, ..., \lambda_M\}$ denote a grid of potential tuning parameters ordered such that $0 < \lambda_1 < \lambda_2 < ... < \lambda_M < \infty$. Fix $\delta \in (0, 1)$. The oracle tuning parameter $\lambda_\delta^*$ is defined as*

$$\lambda_\delta^* = \underset{\lambda \in \Lambda}{\arg\min} \left\{ \mathbb{P}\left\{ \mathcal{A}(\lambda) \right\} \geq 1 - \delta \right\}$$

The oracle tuning parameter provides the tightest upper bound in Proposition 4.1, but is unknowable in practice, since we do not observe $\mathbf{B}^*$ and hence cannot verify $\mathcal{A}$. The aim of our Lepski-type method is to mimic the performance of $\lambda_\delta^*$ in an entirely data-driven fashion. Letting $\Lambda$ denote our ordered grid of potential tuning parameters, we follow [5] and choose the tuning parameter $\hat{\lambda} \in \Lambda$ that satisfies

$$\hat{\lambda} = \underset{\lambda \in \Lambda}{\arg\min} \left\{ \max_{\lambda', \lambda'' \in \Lambda, \lambda', \lambda'' \geq \lambda} \mathcal{P}^*(\nabla \mathcal{L}(\widehat{\mathbf{B}}_{\lambda'}) - \nabla \mathcal{L}(\widehat{\mathbf{B}}_{\lambda''})) \leq \bar{C}(\lambda' + \lambda'') \right\} \quad (5)$$

where $\bar{C}$ is a constant chosen by the statistician. The following theorem states that $\hat{\lambda}$ recovers the behavior of $\lambda_\delta^*$ with high probability, as long as $\bar{C}$ is sufficiently large.

**Theorem 4.1.** *Let $C$ denote the constant in Proposition 4.1. If $\hat{\lambda}$ is chosen as in Equation 5 with $\bar{C} \geq C$, then the following inequalities hold simultaneously with probability at least $1 - \delta$:*

*1. $\hat{\lambda} \leq \lambda_\delta^*$*

*2. $\mathcal{P}^*(\nabla \mathcal{L}(\widehat{\mathbf{B}}_{\hat{\lambda}}) - \nabla \mathcal{L}(\mathbf{B}^*)) \leq C^* \lambda_\delta^*$*

*where $C^* \geq \bar{C}$.*

This theorem is a generalization of Theorem 3 in [5], adapted to our setting. The primary advantage of this Lepski-style tuning scheme is that it can be performed using only the gradient of the loss function, which in our setting consists only of summary-level statistics. This is a marked improvement over other summary statistic-based estimators, which typically require an additional set of individual-level data for tuning.

The adaptive tuning scheme does hold some disadvantages. First of all, it requires a choice of constant $\bar{C}$, which should be taken to be as close to the constant in Proposition 4.1 as possible. Remark 10 in [5] offers some guidance as to how to choose $\bar{C}$ but unfortunately their analysis corresponds only to the LASSO. Deriving the exact constant in Proposition 4.1 may be possible under stronger assumptions on the data-generating process (i.e. Gaussianity), and we view this as an area of future work. Furthermore, Theorem 4.1 offers only a bound on $\nabla \mathcal{L}(\widehat{\mathbf{B}}) - \nabla \mathcal{L}(\mathbf{B}^*)$. Translating this to a bound on $\widehat{\mathbf{B}} - \mathbf{B}^*$ will require strong element-wise conditions on each of the matrices $\widetilde{\mathbf{\Sigma}}^{(q)}$, which we do not explore in the present work. Nevertheless, our simulations in Section 5 indicate that the adaptive tuning method performs well in terms of the MSE of $\widehat{\mathbf{B}}$, suggesting that adaptive tuning is a good option for model selection when only summary statistics are available.

## 5 Numerical experiments and real data application

We validate our theory and demonstrate the effectiveness of multi-task learning in proxy data settings via extensive experiments. In each experiment, we take the proxy dataset to be well-specified;

in other words, we assume that $\Xi = 0$. When $\Xi$ is nonzero, this predictably leads to worse performance, which we demonstrate in the supplement. The code, further implementation details, and additional simulations which explore the use of our adaptive tuning procedure are also available in the supplement.

First, we consider the effect of varying proxy data size on empirical MSE per task. We generate synthetic Gaussian data with $n_{\min} = 100, p = 100, \tilde{n}_{\min} = \tau n_{\min}$ for $\tau \in \{0.5, 1, 2, 5, 10\}$, and $\tilde{\rho}_q = 0$ for each $q$. The number of tasks was fixed at 8. Furthermore, we generate a row-sparse $\mathbf{B}^*$ matrix with 10 nonzero rows and a $\mathbf{B}^*$ with rank 2 for the sparse and low-rank multi-task estimators, respectively. We then fit the proxy data multi-task learning estimator and compare the prediction MSE per task to the estimator that has access to all of the individual level data and to the estimator that uses the true covariance matrix $\Sigma$. The results of this simulation are given in Figure 1.

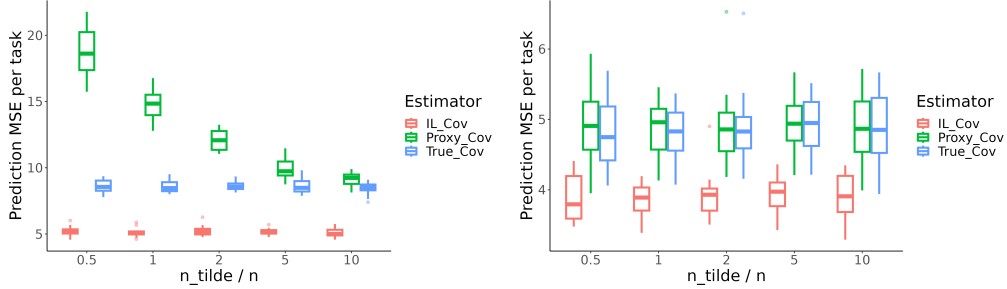

Figure 1: Average prediction MSE per task after 100 repetitions plotted against $\tau = \tilde{n}/n$. The left hand side corresponds to the sparse estimator, and the right hand side is the low-rank estimator. The red boxes indicate the estimator that uses all of the individual-level data (IL_Cov), the blue boxes indicate the estimator that uses the true covariance matrix of the features (true_Cov), and the green boxes correspond to the estimator that uses just the proxy data (Proxy_Cov).

We observe a performance gap between the estimators that use the true covariance matrix and the individual level estimators, as predicted by our theory in Section 3. The performance of the proxy data estimators increase with increasing proxy sample size, but are unable to match the performance of the individual level estimator, as expected.

Next we study the effect of varying the proportion of overlapping samples between the discovery and proxy datasets. Similarly, we generate synthetic data with $n = \tilde{n} = 100$, and vary $\tilde{\rho}$, which indicates the proportion of proxy data points that are also in the discovery dataset. With $Q = 8$, we generate $\mathbf{B}^*$ in the same way as in the previous simulation. These results are given in Figure 2.

Once again, we observe the expected performance gap between the estimators with the true covariance and the individual-level estimators. As the proportion of overlap between the proxy dataset and the discovery dataset grows, we see that the performance of the proxy data estimator converges to that of the individual level estimator. This is anticipated by Theorems 3.1 and 3.2.

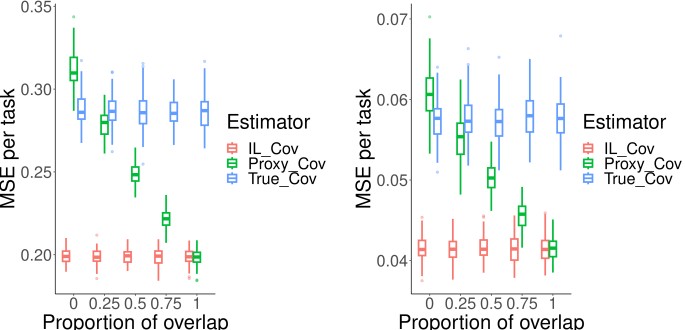

Figure 2: Average MSE per task after 100 repetitions plotted against $\tilde{\rho}$. The orientation and colors are the same as in Figure 1.

Finally, we have applied our method to analyze real genetic data to demonstrate the real-world applicability of our method. We use a multi-site data obtained from the electronic Medical Records and Genomics (eMERGE) network [28], which includes individual-level genotype data from multiple research sites in the United States. Our goal is to predict levels of low-density lipoprotein (LDL) across five adult sites, treating the data from each site as a separate task. We split the data (with sample sizes $n_1 = 3813, n_2 = 546, n_3 = 2666, n_4 = 1435, n_5 = 525$) at each task into a training and test set (with a test set data size of 100 for each task) and evaluate the performance of our method using the prediction MSE on the test set. The training data from each site is used to construct the discovery summary statistics $\widehat{\mathbf{S}}^{(q)}$ for each task. For approximating $\mathbf{\Sigma}^{(q)}$, we choose two different approaches: one is to use the half of the genotype data from each site (this approach is labeled as Proxy_MTL1); the other approach is to use $\mathbf{X}_1$ (genotype data from site 1) to approximate $\mathbf{\Sigma}^{(q)}$ for all the sites. This approach is labeled as Proxy_MTL2. We use these two approaches to demonstrate a potential trade-off in the construction of the reference panel: Proxy_MTL1 uses a well-specified reference dataset with a smaller sample size; Proxy_MTL2 uses a larger reference dataset that may suffer from a distribution shift. For comparison, we also fit a multi-task learning estimator that uses all of the individual level training data for each task, which is labeled 'Individual_MTL', and we fit a ridge regression estimator that models each task separately, for comparison to our multi-task learning approach. The ridge estimator uses the proxy sample covariance instead of the individual level covariance matrix for a fair comparison with our method. We repeat the train-test split process 10 times, which admits the distribution of prediction MSE values that we report in the figure. We use the nuclear norm penalized multi-task learning estimators in this application, because we believe that the genetic effects in this dataset are dense.

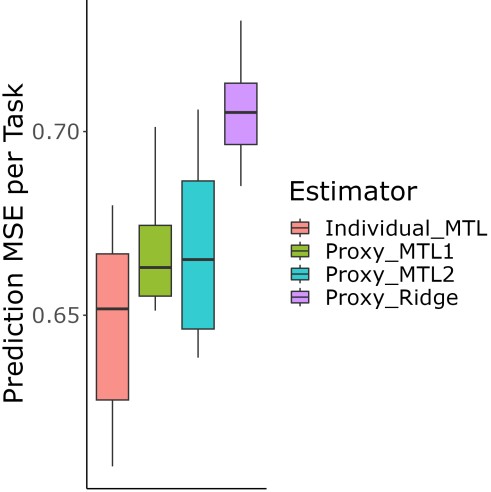

Figure 3: Prediction MSE per task after 10 splits of the eMERGE data.

Our results in Figure demonstrate that our method is highly practical when only summary-level information is available, as the prediction MSE of our method is nearly the same as the estimator which uses the individual-level data, despite a slight cost in performance. Furthermore, all multi-task learning estimators outperform the ridge-estimator, confirming that multi-task learning is a strong approach when there is shared structure between tasks.

## 6 Discussion, Limitations, and Broader Impacts

We have described a flexible multi-task framework incorporating summary statistics from distinct sources with a general data-driven tuning scheme for selecting tuning parameters. Our theoretical analysis sheds light on the intrinsic price of using summary-level information from distinct sources for statistical analysis, and suggests that more overlap between the sources, less distributional shift, and larger proxy data sample sizes can alleviate this cost. Our data-driven tuning scheme allows

models to be trained without sample splitting, making it more applicable to real-world settings with only summary statistics available.

The limitations of our work are summarized as follows. First of all, our methods depend on a linear relationship between the covariates and the outcomes. This assumption is often satisfied in our target application of genetic risk prediction [3], but it does limit the applicability of our method to other domains. To extend our framework to non-linear models, we may use the second-order Taylor approximation of the loss function as in [16]. However, the summary statistics used by such an algorithm are not found in existing literature or publicly available databases. Additionally, our theoretical results provide only upper bounds on the estimation error of the two estimators that we consider in this work. To fully characterize the cost of using summary statistics for multi-task learning, lower bounds resembling Theorem 2.2 of [15] are needed. We conjecture that our estimators converge at a minimax optimal rate, and we view the proof of this conjecture as an important future direction. Finally, we may also extend the framework of [10] to our summary statistic based setting to adjust for potential differences between tasks.

Nevertheless, our results have important implications beyond high-dimensional statistical theory. The trade-off between proxy data sample size and discovery-proxy overlap may inform how polygenic risk models are built in real-world applications: Practitioners should prioritise alignment between the sources of summary statistics that they use to build these models, rather than optimizing for large sample sizes. This guidance may lead to more accurate polygenic scores, which have emerged as an important predictive tool in the field of precision medicine.

We recognize that the development of polygenic risk scores, if done without care, may worsen existing health disparities [23]. This is a potential negative societal impact of our work. We hope that our multi-task learning framework may be used to incorporate data from diverse populations to improve generalizability and transportability of genetic risk predictions to overcome these negative impacts.

## Acknowledgements

Rui Duan is supported by National Institute of General Medical Sciences (NIGM) R01GM148494. Parker Knight is supported by an NSF Graduate Research Fellowship. We would like to thank Rajarshi Mukherjee for an insightful discussion of Lepski's method, and thank the reviewers for their comments and feedback which greatly improved the paper.

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
