# Multi-Task Learning from Summary Statistics: Supplementary Materials

## A Proofs

### A.1 Preliminaries

In this section, we state a handful of standard definitions and concentration results before proving the main theorems.

**Definition A.1.** *A random variable $Z$ is sub-Gaussian with parameter $\nu^2$ if for any $\lambda > 0$, the following inequality holds:*

$$\mathbb{E}\left[e^{\lambda(Z - \mathbb{E}[Z])}\right] \leq e^{\nu^2 \lambda^2 / 2}$$

*We write $Z \in \mathsf{subG}(\nu^2)$*

**Definition A.2.** *A random vector $Z \in \mathbb{R}^d$ is sub-Gaussian with parameter $\nu^2$ if for any constant $a \in \mathbb{R}^d$, we have $\langle Z, a \rangle \in \mathsf{subG}(\|a\|_2^2 \nu^2)$. When this holds, we write $Z \in \mathsf{subG}_d(\nu^2)$.*

**Definition A.3.** *A random variable $W$ is sub-exponential with parameters $\nu^2$ and $\alpha$ if the following inequality holds for all $|\lambda| < \frac{1}{\alpha}$*

$$\mathbb{E}\left[e^{\lambda(W - \mathbb{E}[W])}\right] \leq e^{\nu^2 \lambda^2 / 2}$$

*We write $W \in \mathsf{subE}(\nu^2, \alpha)$*

**Definition A.4.** *A random vector $W \in \mathbb{R}^d$ is sub-exponential with parameters $\nu^2$ and $\alpha$ if for any constant $a \in \mathbb{R}^d$, we have $\langle W, a \rangle \in \mathsf{subE}(\|a\|_2^2 \nu^2, \|a\|_\infty \alpha)$. When this holds, we write $Z \in \mathsf{subE}_d(\nu^2, \alpha)$.*

**Lemma A.1** (Bernstein's inequality). *Let $Z \in \mathsf{subE}(\nu^2, \alpha)$. Then*

$$\mathbb{P}\left\{|Z - \mathbb{E}[Z]| \geq t\right\} \leq 2 \exp\left\{-\min\left(\frac{t^2}{2\nu^2}, \frac{t}{2\alpha}\right)\right\}$$

**Lemma A.2.** *Let $Z \in \mathsf{subE}_d(\nu^2, \alpha)$ with $\mathbb{E}[Z] = 0$. Then there exist constants $C_1$ and $C_2$ such that*

$$\mathbb{P}\left\{\|Z\|_2 \geq t\right\} \leq C_1 \exp\left[C_2(d - \min(t^2/\nu^2, t/\alpha))\right]$$

**Lemma A.3.** *Suppose $\mathbf{A} \in \mathbb{R}^{m \times d}$ is a random matrix whose rows are independent $\mathsf{subE}_d(\nu^2, \alpha)$ random variables with zero mean. Then there exist constants $C_1$ and $C_2$ such that the operator norm of $\mathbf{A}$ satisfies*

$$\mathbb{P}\left\{\|\mathbf{A}\|_{\mathsf{op}} \geq t\right\} \leq C_1 \exp\left[C_2(m + d - \min(t^2/\nu^2, t/\alpha))\right]$$

The proofs of Lemmas A.2 and A.3 follow from covering arguments and Bernstein's inequality; refer to [3] Chapter 4 for details.

## A.2 Proofs of Results in Section 3

The proofs of the upper bounds follow the general structure outlined in [2]. Throughout the proofs, let $\mathcal{L}_q(\mathbf{B}) = \frac{1}{2}\left\|\widetilde{\boldsymbol{\Sigma}}^{(q)1/2}\mathbf{B}e_q\right\|_2^2 - \langle\widehat{\mathbf{S}}^{(q)}, \mathbf{B}e_q\rangle$ and $\mathcal{L}(\mathbf{B}) = \sum_{q\in[Q]}\mathcal{L}_q(\mathbf{B})$. The matrix $\nabla\mathcal{L}(\mathbf{B}^*)$ will play a central role in our analysis; the following lemma gives a characterization of its distribution.

**Lemma A.4.** *There exist constants $C > 0$ and $c > 0$, depending only on $\sigma^2$ and the eigenvalues of the matrices $\boldsymbol{\Sigma}_1^{(q)}$ and $\boldsymbol{\Sigma}_2^{(q)}$, such that the $(i, q)_{th}$ entry of $\nabla\mathcal{L}(\mathbf{B}^*)$ has distribution $\mathsf{subE}(\nu_q^2, \alpha_q)$, where*

$$\nu_q^2 = \frac{C}{n_q}\left(1 + \left\|\beta^{(q)}\right\|_2^2\left(\frac{n_1}{\tilde{n}_q} + 1 - 2\tilde{\rho}_q\right)\right)$$

$$\alpha_q = \left(\frac{c}{n_q \wedge \tilde{n}_q}\right)\left\|\beta^{(q)}\right\|_\infty$$

*Proof.* Direct calculation reveals that the $q_{th}$ column of $\nabla\mathcal{L}(\mathbf{B}^*)$ is equal to $\widehat{\mathbf{S}}^{(q)} - \widetilde{\boldsymbol{\Sigma}}^{(q)}\beta^{(q)}$. Observe

$$\widehat{\mathbf{S}}^{(q)} - \widetilde{\boldsymbol{\Sigma}}^{(q)}\beta^{(q)} = \frac{1}{n_q}\mathbf{X}^{(q)T}\varepsilon^{(q)} + \frac{1}{n_q}\mathbf{X}^{(q)T}\mathbf{X}^{(q)}\beta^{(q)} - \frac{1}{\tilde{n}_q}\widetilde{\mathbf{X}}^{(q)T}\widetilde{\mathbf{X}}^{(q)}\beta^{(q)}$$

$$= \frac{1}{n_q}\mathbf{X}^{(q)T}\varepsilon^{(q)} + \left(\frac{1}{n_q}\mathbf{X}^{(q)T}\mathbf{X}^{(q)} - \frac{1}{\tilde{n}_q}\widetilde{\mathbf{X}}^{(q)T}\widetilde{\mathbf{X}}^{(q)}\right)\beta^{(q)}$$

We clearly have that $\frac{1}{n_q}\mathbf{X}^{(q)T}\varepsilon^{(q)} \in \mathsf{subE}_p(C/n_q, c/n_q)$ for some $C, c > 0$ that depend on $\sigma^2$ and $\boldsymbol{\Sigma}^{(q)}$. To analyze the second term, notice

$$\frac{1}{n_q}\mathbf{X}^{(q)T}\mathbf{X}^{(q)} - \frac{1}{\tilde{n}_q}\widetilde{\mathbf{X}}^{(q)T}\widetilde{\mathbf{X}}^{(q)} = \frac{1}{|\mathcal{I}_q|}\sum_{i\in\mathcal{I}_q}X_iX_i^T - \frac{1}{|\widetilde{\mathcal{I}}_q|}\sum_{i\in\widetilde{\mathcal{I}}_q}X_iX_i^T$$

$$= \sum_{i\in\mathcal{I}_q\cap\widetilde{\mathcal{I}}_q}\left(\frac{1}{|\mathcal{I}_q|} - \frac{1}{|\widetilde{\mathcal{I}}_q|}\right)X_iX_i^T$$

$$+ \sum_{i\in\mathcal{I}_q-\widetilde{\mathcal{I}}_q}\frac{1}{|\mathcal{I}_q|}X_iX_i^T$$

$$- \sum_{i\in\widetilde{\mathcal{I}}_q-\mathcal{I}_q}\frac{1}{|\widetilde{\mathcal{I}}_q|}X_iX_i^T$$

$$= \mathrm{I} + \mathrm{II} - \mathrm{III}$$

Right-multiplying by $\beta^{(q)}$ and applying standard properties of sub-exponential random variables, we get

$$\mathrm{I}\beta^{(q)} \in \mathsf{subE}_p\left(C_1|\mathcal{I}_q\cap\widetilde{\mathcal{I}}_q|\left[\frac{1}{|\mathcal{I}_q|} - \frac{1}{|\widetilde{\mathcal{I}}_q|}\right]^2\left\|\beta^{(q)}\right\|_2^2, c_1\left[\frac{1}{|\mathcal{I}_q|} - \frac{1}{|\widetilde{\mathcal{I}}_q|}\right]\left\|\beta^{(q)}\right\|_\infty\right)$$

$$\mathrm{II}\beta^{(q)} \in \mathsf{subE}_p\left(C_2\frac{|\mathcal{I}_q-\widetilde{\mathcal{I}}_q|}{|\mathcal{I}_q|^2}\left\|\beta^{(q)}\right\|_2^2, \frac{c_2}{|\mathcal{I}_q|}\left\|\beta^{(q)}\right\|_\infty\right)$$

$$\mathrm{III}\beta^{(q)} \in \mathsf{subE}_p\left(C_3\frac{|\widetilde{\mathcal{I}}_q-\mathcal{I}_q|}{|\widetilde{\mathcal{I}}_q|^2}\left\|\beta^{(q)}\right\|_2^2, \frac{c_3}{|\widetilde{\mathcal{I}}_q|}\left\|\beta^{(q)}\right\|_\infty\right)$$

To find the distribution of $(\text{I} + \text{II} - \text{III})\beta^{(q)}$, we take the sum of the first parameters and the max of the second parameters, and choose constants $C_4 = \max(C_1, C_2, C_3)$ and $c_4 = \max(c_1, c_2, c_3)$. After some arithmetic this yields

$$(\text{I} + \text{II} - \text{III})\beta^{(q)} \in \mathsf{subE}_p\left(C_4\left(\frac{1}{\tilde{n}_q} - 2\frac{|\mathcal{I}_q \cap \widetilde{\mathcal{I}}_q|}{n_q \tilde{n}_q} + \frac{1}{n_q}\right)\left\|\beta^{(q)}\right\|_2^2, \frac{c_4}{n_q \wedge \tilde{n}_q}\left\|\beta^{(q)}\right\|_\infty\right)$$

where we use $|\mathcal{I}_q| = n_q$ and $|\widetilde{\mathcal{I}}_q| = \tilde{n}_q$ for clarity, as well as $1/x - 1/y \leq \max(1/x, 1/y)$ for $x, y > 0$ to simplify the second parameter. Combining this with $\frac{1}{n_q}\mathbf{X}^{(q)T}\varepsilon^{(q)} \in \mathsf{subE}_p(C/n_q, c/n_q)$, we have that

$$\widehat{\mathbf{S}}^{(q)} - \widetilde{\mathbf{\Sigma}}^{(q)}\beta^{(q)} \in \mathsf{subE}_p\left(C\left[\frac{1}{n_q} + \left(\frac{1}{\tilde{n}_q} - 2\frac{|\mathcal{I}_q \cap \widetilde{\mathcal{I}}_q|}{n_q \tilde{n}_q} + \frac{1}{n_q}\right)\left\|\beta^{(q)}\right\|_2^2\right], \frac{c}{n_q \wedge \tilde{n}_q}\left\|\beta^{(q)}\right\|_\infty\right)$$

for $C, c$ sufficiently large. This completes the proof.

$\square$

### A.2.1 Proof of Theorem 3.1

*Proof.* Let $\widehat{\mathbf{\Delta}} = \widehat{\mathbf{B}}^{(sp)} - \mathbf{B}^*$, and define the events

$$\mathcal{A}_1 = \left\{\frac{\lambda}{2} \geq \|\nabla\mathcal{L}(\mathbf{B}^*)\|_{2,\infty}\right\}$$

$$\mathcal{A}_2 = \left\{\sum_{q=1}^{Q}\left\|\widetilde{\mathbf{\Sigma}}^{(q)1/2}\Delta e_t\right\|_2^2 \geq \frac{1}{\kappa}\|\Delta\|_F^2 \quad \forall\Delta \in \mathcal{C}_3(S^*)\right\}$$

The following analysis is conditional on $\mathcal{A}_1 \cap \mathcal{A}_2$.

We first need that $\widehat{\mathbf{\Delta}} \in \mathcal{C}_3(S^*)$. This is a standard result for high-dimensional M-estimators and we omit the proof for brevity. See Proposition 9.13 in [4].

By the optimality of $\widehat{\mathbf{B}}^{(sp)}$, we know that

$$\sum_{q=1}^{Q}\left\|\widetilde{\mathbf{\Sigma}}^{(q)1/2}\widehat{\mathbf{\Delta}}\right\|_2^2 \lesssim \underbrace{\sum_{q=Q}^{T}\langle\widehat{\mathbf{S}}^{(q)} - \widetilde{\mathbf{\Sigma}}^{(q)}\mathbf{B}^*e_q, \widehat{\mathbf{\Delta}}e_q\rangle}_{\text{I}} - \underbrace{\lambda(\left\|\widehat{\mathbf{B}}^{(sp)}\right\|_{2,1} - \|\mathbf{B}^*\|_{2,1})}_{\text{II}}$$

Recall that $\nabla\mathcal{L}(\mathbf{B}^*)$ has its $q_{th}$ column equal to $\widehat{\mathbf{S}}^{(q)} - \widetilde{\mathbf{\Sigma}}^{(q)}\mathbf{B}^*e_q$. So we can rewrite I as

$$\text{I} = \langle\nabla\mathcal{L}(\mathbf{B}^*), \widehat{\mathbf{\Delta}}\rangle \leq \|\nabla\mathcal{L}(\mathbf{B}^*)\|_{2,\infty}\left\|\widehat{\mathbf{\Delta}}\right\|_{2,1} \lesssim \lambda\sqrt{s}\left\|\widehat{\mathbf{\Delta}}\right\|_F$$

where the first inequality is Holder's, and the second uses the fact that we are conditioned on $\mathcal{A}_1$ and $\|\Delta\|_{2,1} \lesssim \sqrt{s}\|\Delta\|_F$ for $\Delta \in \mathcal{C}_\alpha(S)$. To control II, we apply Lemma 9.14 in [4] and conclude

$$\text{II} \lesssim \lambda\sqrt{s}\left\|\widehat{\mathbf{\Delta}}\right\|_F$$

which gives

$$\sum_{q=1}^{Q} \left\| \widetilde{\boldsymbol{\Sigma}}^{(q)\,1/2} \widehat{\boldsymbol{\Delta}} e_q \right\|_2^2 \lesssim \lambda \sqrt{s} \left\| \widehat{\boldsymbol{\Delta}} \right\|_F$$

Since we are conditioned on $\mathcal{A}_2$ as well, the left hand side of the above inequality is bounded below by $\frac{1}{\kappa} \left\| \widehat{\boldsymbol{\Delta}} \right\|_F^2$. This yields

$$\left\| \widehat{\mathbf{B}}^{(sp)} - \mathbf{B}^* \right\|_F \lesssim \lambda \sqrt{s}$$

For our choice of $\lambda = O(\gamma(Q + \log p)/n_{\min} + \|\Xi\|_{2,\infty})$, we have

$$\left\| \widehat{\mathbf{B}}^{(sp)} - \mathbf{B}^* \right\|_F \lesssim \sqrt{\frac{s\gamma(Q + \log p)}{n_{\min}}} + \sqrt{s} \|\Xi\|_{2,\infty}$$

It remains to show that $\mathcal{A}_1 \cap \mathcal{A}_2$ occurs with high probability. We know by Assumption 3.3 that $\mathcal{A}_2$ occurs with probability at least $1 - a_N$. To analyze $\mathcal{A}_1$, recall that Lemma A.4 tells us us that the $(i,q)_{th}$ entry of $\nabla \mathcal{L}(\mathbf{B}^*)$ has a $\mathsf{subE}(v_q^2, \alpha_q)$ distribution. Letting $\ell_i$ denote the $i_{th}$ row of $\nabla \mathcal{L}(\mathbf{B}^*)$, it follows that $\ell_i \in \mathsf{subE}_Q(v^2, \alpha)$ for $v^2 = \max_q v_q^2$ and $\alpha = \max_q \alpha_q$. Using this, an application of Lemma A.2 and the union bound gives us

$$\begin{aligned}
\mathbb{P}\left\{ \|\nabla \mathcal{L}(\mathbf{B}^*) - \mathbb{E}\left[\nabla \mathcal{L}(\mathbf{B}^*)\right]\|_{2,\infty} \geq t \right\} &= \mathbb{P}\left\{ \max_{i \in [p]} \|\ell_i - \mathbb{E}\left[\ell_i\right]\|_2 \geq t \right\} \\
&\leq \sum_{i=1}^{p} \mathbb{P}\left\{ \|\ell_i - \mathbb{E}\left[\ell_i\right]\|_2 \geq t \right\} \\
&\leq \sum_{i=1}^{p} C_1 \exp[C_2(Q - \min(t^2/\nu^2, t/\alpha))] \\
&= \exp[C(\log p + Q - \min(t^2/\nu^2, t/\alpha))]
\end{aligned}$$

A direct calculation reveals that $\mathbb{E}\left[\nabla \mathcal{L}(\mathbf{B}^*)\right] = \Xi$. So, using our choice $t = \lambda = C(\sqrt{\gamma(Q + \log p)/n_{\min}} + \|\Xi\|_{2,\infty})$ yields the desired result, as long as $n_{\min} \wedge \tilde{n}_{\min} \geq c \|\mathbf{B}^*\|_{\infty,\infty} (Q + \log p)$.

$\square$

### A.2.2 Proof of Theorem 3.2

*Proof.* Define the events

$$\mathcal{B}_1 = \left\{ \frac{\lambda}{2} \geq \|\nabla \mathcal{L}(\mathbf{B}^*)\|_{\mathsf{op}} \right\}$$

$$\mathcal{B}_2 = \left\{ \sum_{t=q}^{Q} \left\| \widetilde{\boldsymbol{\Sigma}}^{(q)\,1/2} \Delta e_q \right\|_2^2 \geq \frac{1}{\kappa} \|\Delta\|_F^2 \quad \forall \Delta \in \mathcal{C}_3(\mathbb{M}^*) \right\}$$

Conditional on $\mathcal{B}_1 \cap \mathcal{B}_2$, an analysis identical to that given in the proof of Theorem 3.1 grants us

$$\left\| \widehat{\mathbf{B}}^{(lr)} - \mathbf{B}^* \right\|_F \lesssim \lambda \sqrt{r}$$

which admits

$$\left\|\widehat{\mathbf{B}}^{(lr)} - \mathbf{B}^*\right\|_F \lesssim \sqrt{\frac{r\xi(T+p)}{n_{\min}}} + \sqrt{r}\,\|\Xi\|_{\mathsf{op}}$$

due to our choice of $\lambda$. By assumption 3.5, we know that $\mathcal{B}_2$ occurs with probability at least $1 - b_N$. The fact that $\mathcal{B}_1$ holds with high probability follows directly from Lemmas A.4 and A.3.

$\square$

### A.3  Proofs of Results in Section 4

#### A.3.1  Proof of Proposition 4.1

*Proof.* Since $\mathcal{P}$ is convex, the estimator $\widehat{\mathbf{B}}$ satisfies the following first-order condition:

$$\nabla\mathcal{L}(\widehat{\mathbf{B}}) + \lambda\widehat{\mathbf{Z}} = 0$$

where $\widehat{\mathbf{Z}}$ lies in the sub-gradient of $\mathcal{P}$ at $\widehat{\mathbf{B}}$. Subtracting $\nabla L(\mathbf{B}^*)$ from both sides and rearranging terms grants us

$$\nabla\mathcal{L}(\widehat{\mathbf{B}}) - \nabla\mathcal{L}(\mathbf{B}^*) = -\lambda\widehat{\mathbf{Z}} - \nabla\mathcal{L}(\mathbf{B}^*)$$

Applying $\mathcal{P}^*$ to both sides, we get

$$\begin{aligned}
\mathcal{P}^*(\nabla\mathcal{L}(\widehat{\mathbf{B}}) - \nabla\mathcal{L}(\mathbf{B}^*)) &= \mathcal{P}^*(-\lambda\widehat{\mathbf{Z}} - \nabla\mathcal{L}(\mathbf{B}^*)) \\
&\leq \lambda\mathcal{P}^*(\widehat{\mathbf{Z}}) + \mathcal{P}^*(\nabla\mathcal{L}(\mathbf{B}^*)) \\
&\leq \lambda + \frac{\lambda}{2}
\end{aligned}$$

where the first inequality uses the triangle inequality, and the second uses the properties of sub-gradients and the event $\mathcal{A}(\lambda)$.

$\square$

#### A.3.2  Proof of Theorem 4.1

This basically follows the proof of Theorem 1 in [1]. We repeat the proof for completeness.

*Proof.* Condition on $\mathcal{A}(\lambda_\delta^*)$.

We first prove that $\hat{\lambda} \leq \lambda_\delta^*$. We proceed by contradiction: suppose that $\hat{\lambda} > \lambda_\delta^*$. Then by the definition of $\hat{\lambda}$, there exist $\lambda', \lambda'' \geq \lambda_\delta^*$ such that

$$\mathcal{P}^*(\nabla\mathcal{L}(\widehat{\mathbf{B}}_{\lambda'}) - \nabla\mathcal{L}(\widehat{\mathbf{B}}_{\lambda''})) > C(\lambda' + \lambda'')$$

Since $\mathcal{A}(\lambda')$ and $\mathcal{A}(\lambda'')$ are subsets of $\mathcal{A}(\lambda_\delta^*)$, Proposition 4.1 tells us that the following inequalities hold:

$$\begin{aligned}
\mathcal{P}^*(\nabla\mathcal{L}(\widehat{\mathbf{B}}_{\lambda'}) - \nabla\mathcal{L}(\mathbf{B}^*)) &\leq C\lambda' \\
\mathcal{P}^*(\nabla\mathcal{L}(\widehat{\mathbf{B}}_{\lambda''}) - \nabla\mathcal{L}(\mathbf{B}^*)) &\leq C\lambda''
\end{aligned}$$

So we have

$$\begin{aligned}
\mathcal{P}^*(\nabla\mathcal{L}(\widehat{\mathbf{B}}_{\lambda'}) - \nabla\mathcal{L}(\widehat{\mathbf{B}}_{\lambda''})) &\leq \mathcal{P}^*(\nabla\mathcal{L}(\widehat{\mathbf{B}}_{\lambda'}) - \nabla\mathcal{L}(\mathbf{B}^*)) + \mathcal{P}^*(\nabla\mathcal{L}(\widehat{\mathbf{B}}_{\lambda''}) - \nabla\mathcal{L}(\mathbf{B}^*)) \\
&\leq C(\lambda' + \lambda'')
\end{aligned}$$

But $\bar{C} \geq C$, so this is a contradiction. Hence $\hat{\lambda} \leq \lambda_\delta^*$.

Now we prove the second claim. Since we are still conditioned on $\mathcal{A}(\lambda_\delta^*)$, we know that $\hat{\lambda} \leq \lambda_\delta^*$. So applying the definition of $\hat{\lambda}$, we have

$$\mathcal{P}^*(\nabla\mathcal{L}(\widehat{\mathbf{B}}_{\hat{\lambda}}) - \nabla\mathcal{L}(\widehat{\mathbf{B}}_{\lambda_\delta^*})) \leq \bar{C}(\hat{\lambda} + \lambda_\delta^*) \leq 2\bar{C}\lambda_\delta^*$$

Using this result, we apply the triangle inequality to yield

$$\begin{aligned}
\mathcal{P}^*(\nabla\mathcal{L}(\widehat{\mathbf{B}}_{\hat{\lambda}}) - \nabla\mathcal{L}(\mathbf{B}^*)) &\leq \mathcal{P}^*(\nabla\mathcal{L}(\widehat{\mathbf{B}}_{\hat{\lambda}}) - \nabla\mathcal{L}(\widehat{\mathbf{B}}_{\lambda_\delta^*})) + \mathcal{P}^*(\nabla\mathcal{L}(\widehat{\mathbf{B}}_{\lambda_\delta^*}) - \nabla\mathcal{L}(\mathbf{B}^*)) \\
&\leq 2\bar{C}\lambda_\delta^* + C\lambda_\delta^* \\
&\leq C^*\lambda_\delta^*
\end{aligned}$$

where $C^* \geq \bar{C}$.

Since $\mathcal{A}(\lambda_\delta^*)$ occurs with probability at least $1 - \delta$, this completes the proof.

$\square$

## B   Simulations

All simulations, including those in the main text, were run in R version 4.0.2 on a Linux machine with an Intel i5 processor. We implemented all of the estimators using a straightforward proximal gradient descent algorithm with step size fixed at 1e-3.

For our first additional simulation, we compare the multi-task learning estimators to analogous single-task estimators. Specifically, we compare the sparse multi-task estimator defined in the main text to the LASSO computed using only proxy data, and we compare the low-rank estimator to Ridge regression also computed with proxy data. We generate synthetic Gaussian data with $n_{\min} = 100, \tilde{n}_{\min} = 150, p = 100$ and $\tilde{\rho} = 0$ so that there is no overlap between summary statistics. For the sparse estimators, we generate the $\mathbf{B}^*$ matrix with 10 nonzero rows, and for the low-rank estimators, we generate $\mathbf{B}^*$ with a rank of 2. For both comparisons, we generate the columns of $\mathbf{B}^*$ distinctly to model heterogeneity between tasks. Finally, we consider two versions of the LASSO/Ridge estimators. The first variation pools all of the data across tasks into one dataset, and computes an estimate $\hat{\beta}$. This approach completely ignores heterogeneity between tasks. The second variation models each task separately without consider structural similarities between tasks. We choose these two variations to demonstrate the effectiveness of multi-task learning in situations when tasks should be modeled separately, but there is some shared structure between them.

The results of this simulation are shown in Figure 1. Clearly the multi-task estimators outperform the single-task analogs, which is expected given the data generating process. This suggests that multi-task learning should be preferred for integrating data from similar but distinct sources.

Next, we consider the effect of a misspecified proxy data set on the MSE per task. This simulation is intended to study the effect of distributional shifts between the $\mathbf{X}^{(q)}$ and $\widetilde{\mathbf{X}}^{(q)}$ matrices for each task on the downstream performance of our estimators. This simulation setup is the same as the last one, except we draw $\mathbf{X}^{(q)}$ from a normal distribution with covariance $\Sigma_1$ and $\widetilde{\mathbf{X}}^{(q)}$ from a normal distribution with covariance $\Sigma_2$, where we vary $\|\Sigma_1 - \Sigma_2\|_F \in \{10; 20; 50; 100\}$.

The results of this simulation are found in Figure 2. We can clearly see that the proxy data should be well-specified to decrease the MSE per task, as expected.

Finally, we compare our adaptive tuning procedure to a hold-out validation procedure that uses a small amount of individual-level data each task. The hold-out validation scheme assumes that we have access to a dataset $(X_{\text{tune}}^{(q)}, Y_{\text{tune}}^{(q)})$ for each task $q \in [Q]$, and chooses $\lambda \in \Lambda$ such that it minimizes $\sum_{q=1}^{Q} \left\| Y_{\text{tune}}^{(q)} - X_{\text{tune}}^{(q)}\widehat{\mathbf{B}}_\lambda e_q \right\|_2^2$, where $\widehat{\mathbf{B}}_\lambda$ is computed using the proxy data set which is independent from $(X_{\text{tune}}^{(q)}, Y_{\text{tune}}^{(q)})_{q \in [Q]}$. This hold-out tuning procedure is often used in practice, especially in statistical genetics, whenever such a dataset is available. However, when it comes to multi-task

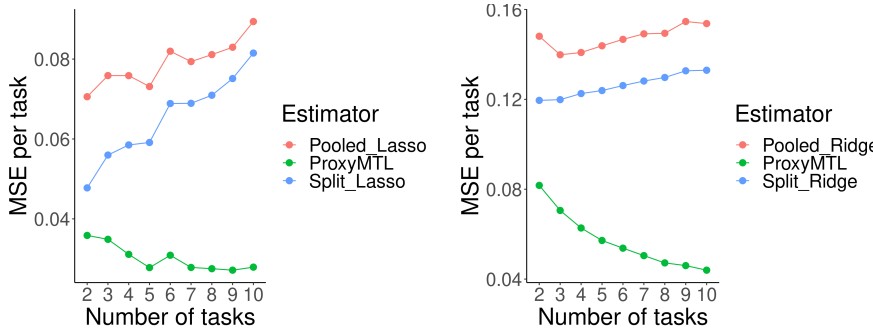

Figure 1: Average MSE per task after 100 repetitions plotted against the number of tasks. The plot on the left-hand side corresponds to the sparse estimator, and the figure on the right is the low-rank estimator. In both plots, the 'ProxyMTL' line corresponds to our multi-task learning method. The 'Pooled Lasso' and 'Pooled Ridge' correspond to the estimators fit by pooling all the data across the tasks, and the 'Split Lasso' and 'Split Ridge' correspond to estimators fit on each task separately. The MSE per task is computed by summing up the MSE accumulated across all of the tasks, then dividing by the number of tasks.

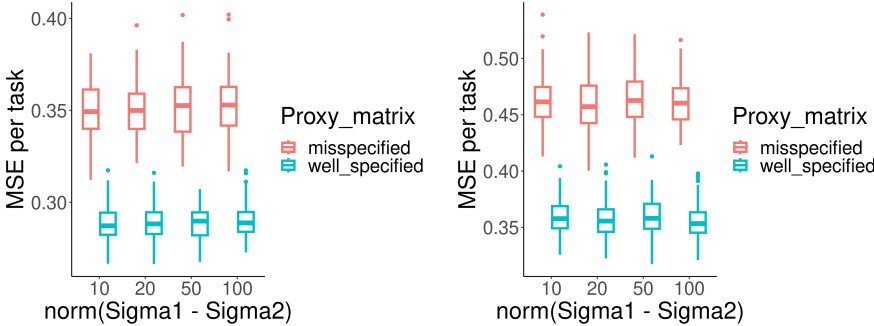

Figure 2: Average MSE per task after 100 repetitions plotted against $\|\Sigma_1 - \Sigma_2\|_F$. The left plot corresponds to the sparse estimator and right plot corresponds to the low-rank estimator.

learning, obtaining validation data for all $Q$ tasks can pose a significant challenge. Fortunately, our adaptive tuning procedure provides a compelling alternative that overcomes this obstacle.

We present the results of our simulations in Figure 3. In these simulations, we vary the sample size of the hold-out dataset from 10 to 100. The y-axis is the average MSE per task of the estimator computed using the tuning parameter chosen by each of the two methods. Furthermore, we have pooled the hold-out data with the proxy data in computing the estimator with the adaptive validation method, to emphasize that adaptive validation is able to take full advantage of the data at hand without needing an additional set of tuning data. This adaptive method offers comparable performance to hold-out tuning, since pooling the data increases sample size as well as overlap between $\widehat{\mathbf{S}}^{(q)}$ and $\widetilde{\mathbf{\Sigma}}^{(q)}$ for each $q$. The performance of adaptive tuning improves as the amount of hold-out data increases, as expected.

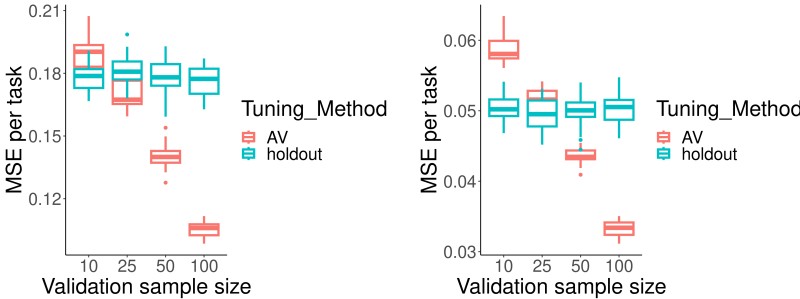

Figure 3: Average MSE per task after 100 repetitions plotted against the choice of tuning method. 'AV' standards for adaptive validation, which refers to our method outlined in the main manuscript with $\bar{C} = 1$. The label 'holdout' refers to the hold-out validation method outlined above. The figure on the left-hand side gives the results for the sparse estimator, and the low-rank estimator is on the right.