# OpenReview forum: "Multi-task learning with summary statistics"
_NeurIPS.cc/2023/Conference — NeurIPS 2023 poster_

### Official Review · Reviewer_wP1d · 2023-06-27

**Soundness:** 3 good
**Presentation:** 4 excellent
**Contribution:** 2 fair
**Rating:** 6
**Confidence:** 3

**Summary:**

The work considers multi-task learning in settings where for each task only summary statistics X.T@Y and ẍ.T@ẍ are made available. The setting is motivated by healthcare and biomedical related research, where sharing individual level microdata is restricted by regulations due to privacy concerns. The authors assume a linear and sparse or low-rank underlying model. A regularized least-squares type of optimization framework is used, and it is highlighted that only summary statistics are needed to solve it. A main contribution of the paper is theoretical analysis bounding the the quality of the sparse or low-rank estimator taking into account quantities such as overlap and distributional shift between the so called discovery and proxy data used to calculate the summary statistics. Additionally, the problem of hyperparameter tuning is considered as basic (cross-)validation methods are not applicable without access to individual data, and Lepski’s method is proposed as a practical alternative that requires only summary level data as well as some prior knowledge in form of a constant C parameter. Experiments on simulated data show results that are consistent with the theoretical analysis.

**Strengths:**

- In general, approaches that allow learning from summary statistics are becoming ever more relevant to the community as regulation on sharing individual level data is tightening
- The exact formulation of the multi-task summary statistic learning problem considered in the paper, where separate discovery and proxy data sets are assumed appears to be novel, as consequently the theoretical bounds that take this into account provide also novel insights about the effects this can have on learning.
- Both L1,2 and nuclear norm regularization based variants of the learning problem are analyzed
- The proposed approach for model selection provides a practical tool for hyperparameter tuning when only summary statistics are available, and might have implications also for other variations of the problem setting?
- Fairly clear writing and technically rigorous

**Weaknesses:**

- Real-world relevance: while applications such as polygenic risk prediction are mentioned, there are no convincing examples, real applications of benchmark data sets provided where the assumptions of the learning setting (having related tasks, only X(q).T@Y(q) and ẍ(q).T@ẍ(q) available, perhaps X(q) != ẍ(q)) would hold. Thus the question about practical significance of the exact considered problem setting and proposed solution remain unclear.
- Related to above point - no experiments on real-world data that would demonstrate the benefits of the approach
- Somewhat restrictive assumptions such as assumptions about the linearity of the underlying model and relationships between the tasks

**Questions:**

Can you clarify on the assumption that the discovery and proxy data sets can be separate, when would this in practice be an important consideration?
Figure 3: am I reading this correctly that the holdout method does not benefit at all from having larger validation sample size? Any explanation for this?


**Limitations:**

-

---

> ### Author Rebuttal · Authors · 2023-08-10
>
> Regarding the real-world application: To address this concern, we have included a real data application to polygenic risk prediction. Details of our analysis and the results are provided in the global response.
>
> **Q1** In statistical genetics applications, many studies that investigate the marginal relationships between covariates and the outcome will report $X^TY$ but do not report $X^TX$. For this reason, the sample covariance must be computed from an external reference data set, such as the 1000 Genomes data [1]. The formulation of our problem and the theory that we develop captures this discrepancy between the datasets.
>
> **Q2** The holdout method does not benefit from a larger sample size because the estimation accuracy will depend only on the sample size of the training dataset, as long as the tuning parameter is chosen to be the right order of magnitude. For the results in Figure 3, the grid of tuning parameters over which the holdout validation was performed was specified to be roughly the appropriate order.
>
> [1] The 1000 Genomes Project Consortium et al., “A global reference for human genetic variation,” Nature, vol. 526, no. 7571, pp. 68–74, Oct. 2015, doi: 10.1038/nature15393.

---

> > ### Comment · Reviewer_wP1d · 2023-08-14
> >
> > Thank you for the addition of the real-world application and for the clarifications provided. I agree that these improve the quality of the work and have raised my rating accordingly to "6: weak accept".

---

### Official Review · Reviewer_Z4Yp · 2023-07-05

**Soundness:** 3 good
**Presentation:** 3 good
**Contribution:** 3 good
**Rating:** 7
**Confidence:** 2

**Summary:**

The paper proposed a multi-task method to learn individual models without having access to the raw data but using summary statistics data for each task. The method is only applicable to linear models.

**Strengths:**

1.  Theoretical guarantee for the optimal estimator
2.  Good experiments on simulated data

**Weaknesses:**

1. Missing experiments on real data
2. Applicable only to linear models.

**Questions:**

1. Why is the discovery data estimated from the actual observed data, unlike the sample covariance matrix was estimated from a proxy data?
2. The proposed method works only for linear models. Can the same analysis be applied using non-linear (deep learning) models?
3. In Figure 1, can you explain the behavior that the proxy covariance gave lower MSE than the tru covariance?

**Limitations:**

Yes.

---

> ### Author Rebuttal · Authors · 2023-08-10
>
> Regarding experiments on real data: We have included additional results demonstrating our method on a prediction task with real genetic data. We have provided a description of the analysis and the results in our global response.
>
> **Q1** In statistical genetics applications, it is often the case that $X^TY$ is reported from studies that analyze the marginal associations between the covariates, which often correspond to genetic markers, and the outcome of interest. However, these studies rarely report the covariance structure between genetic markers, and this information needs to be obtained from additional external datasets, such as the 1000 Genomes data [1]. Our formulation captures this discrepancy between the data used to compute the marginal associations and the covariance matrix.
>
> **Q2** Our results are tailored towards linear models, but our approach may be applied to any estimator whose loss function depends only on the sample covariance matrix and summary association statistics. We anticipate that similar results may be possible for other classes of (non-linear) models such as deep neural networks. However, to fit a neural network using only summary-level information, practitioners need to be able to share new classes of summary statistics that correspond to more complex loss functions. The problem could be formulated as a federated learning problem [2], in which data owners iteratively share gradient-type information for model updating. However, this approach differs from our motivation, which aims to leverage pre-existing summary stats instead of requiring continuous sharing during model training.
>
> **Q3** The behavior in Figure 1 is due to a pre-processing step in the simulation that regularizes the proxy data for better numerical stability across replications. If this pre-processing step is not performed, the behavior of the proxy data estimator converges to that of the true covariance, as predicted by our theory. In the new simulation result described in our global response, we do not perform this pre-processing step, and the estimator behaves as expected according to our theory.
>
> [1] The 1000 Genomes Project Consortium et al., “A global reference for human genetic variation,” Nature, vol. 526, no. 7571, pp. 68–74, Oct. 2015, doi: 10.1038/nature15393.
> [2] M. I. Jordan, J. D. Lee, and Y. Yang, “Communication-Efficient Distributed Statistical Inference,” Journal of the American Statistical Association, vol. 114, no. 526, pp. 668–681, Apr. 2019, doi: 10.1080/01621459.2018.1429274.

---

### Official Review · Reviewer_TqYc · 2023-07-07

**Soundness:** 3 good
**Presentation:** 3 good
**Contribution:** 2 fair
**Rating:** 5
**Confidence:** 4

**Summary:**

This paper presents an approach to learning predictive models from summary statistics in the setting of multi-task learning.  Linear model is assumed and least square solution was derived with either the \ell_{2,1}-norm or kernel norm of the parameter matrix to capture relativeness among tasks. Theoretical results for bounding the estimator when proxy data are used to obtain the covariance matrix and hyperparameter tunning are provided. Synthetic data was used in their empirical evaluation.

**Strengths:**

+ The studied problems, both training predictive models from summary statistics and the use of proxy data to estimate covariance are of great importance, given the constraints on data sharing in medical research and the lack of availability of covariance in current practice of GWAS.
+ Theoretical results can be very useful in assessing the impact of the use of proxy data.
+ The approach proposed to tune the hyperparameters is important given there is typically no validation set consisting of individual level data for such tunning.

**Weaknesses:**

- Even though the studied problems are significant, the proposed approach lacks technical innovations, seeming trivial extensions of existing works for both the bound derivation and hyperparameter tunning. The authors may point out the technique challenges in these extensions, highlighting their technical contribution.
- Lack of evaluation on practical datasets. It is unclear how useful this method is in practical problems.

**Questions:**

- No test set is mentioned in the description of their experiment. Is there a separate test set? I do not think training MSE is a good measure as it is affected by the sparse driving term in the optimization.
-  I found the results shown in Figure 2 is a bit confusing. Given n = \tilde{n}, \tilde{p} = 1 means the proxy data is exactly the true data, meaning the proxy covariance is exactly the true covariance. Also, covariance is the only input from the proxy data to obtain the solution according to the estimator. These together mean the solution obtained with proxy_cov should be the same as that with true_cov at \tilde{p}=1. How the two plots show the former is the same as that with IL_cov?

---

> ### Author Rebuttal · Authors · 2023-08-10
>
> We thank the reviewer for the valuable feedback, and for recognizing the strengths of our paper and the importance of the problem that we address.
>
> Regarding the technical innovation of our paper: The proofs of the theoretical results present unique challenges in the proxy data setting. The derivation of the error bounds relies on a careful analysis of the contributed variance from each data point $X_i$ and $\tilde{X}_i$, which to our knowledge is entirely novel for high dimensional linear models. The details of this analysis can be found in the proof of Lemma A.4 in our supplement. The closest related result to ours is Theorem 2.1 of [2], but our results generalize this theorem by accounting for the overlap between the reference data and the discovery data and by allowing for multiple outcomes. Our derivations for the hyperparameter tuning procedure are generalizations of the techniques used in [1]. To our knowledge, this is the first application of Lepski’s method to high-dimensional regression problems beyond the Lasso and is the first time that the use of Lepski’s method has been motivated by data-sharing restrictions. Our proofs for the model tuning procedure rely on the properties of general convex regularizers, analogous to the framework provided in [3].
>
> Regarding evaluation on practical datasets: Thank you for this suggestion. We have performed a data analysis demonstrating the performance of our method on a real genetic dataset, where the goal is to predict low-density lipoprotein (LDL) levels using genotype data. The results and an in-depth description of the analysis are given in our global response.
>
> **Q1** Our previous simulation results present the mean-squared error of parameter estimation per task; in other words, we present the quantity $\frac{1}{nQ}\|\|\hat{B} - B^*\|\|^2$. We use the entire dataset to compute $\hat{B}$, and so we did not use a test set. We will clarify this in the final version of our manuscript. To address your concerns, our new real data application uses a standard train-test split to evaluate the prediction performance. We provide more details in the global response. Additionally, we have performed an additional simulation study that analyzes the impact of the proxy data sample size on prediction performance for each of our estimators, under the regime of no-overlap between the proxy and discovery data. The details and results of this simulation are in the additional PDF provided in the global response.
>
> **Q2** When $\rho = 1$, the proxy data is the same as the individual-level data, meaning that $\hat{\Sigma} = \tilde{\Sigma}$. In this case, the Proxy_Cov estimator is expected to perform as well as the IL_Cov estimator, which is confirmed by our simulation results in Figure 2. The ‘true’ covariance label refers to the underlying population-level covariance which we do not observe in practice. The True_Cov estimator uses this population-level matrix as the input to our estimator. This is not possible in practice, but we include the results on our figure to demonstrate that overlap between the proxy data and discovery data is more important for good statistical performance than having an increasingly large reference panel.
>
>
> [1] M. Chichignoud, J. Lederer, and M. Wainwright, “A Practical Scheme and Fast Algorithm to Tune the Lasso With Optimality Guarantees.” arXiv, Nov. 08, 2016. Accessed: Apr. 24, 2023. [Online]. Available: http://arxiv.org/abs/1410.0247
> [2] S. Li, T. T. Cai, and H. Li, “Estimation and Inference with Proxy Data and its Genetic Applications,” arXiv:2201.03727 [math, stat], Jan. 2022, Accessed: Mar. 07, 2022. [Online]. Available: http://arxiv.org/abs/2201.03727
> [3] S. N. Negahban, P. Ravikumar, M. J. Wainwright, and B. Yu, “A Unified Framework for High-Dimensional Analysis of $M$-Estimators with Decomposable Regularizers,” Statist. Sci., vol. 27, no. 4, Nov. 2012, doi: 10.1214/12-STS400.

---

> > ### Comment · Reviewer_TqYc · 2023-08-11
> > **Re: Rebuttal by Authors**
> >
> > Thanks to the authors for taking time to respond to my comments. More detailed description of synthetic data generation is desired. What's the role of true (population) \Sigma in this process? How was it specified/computed? Any thoughts on why the use of true \Sigma led to poorer results?

---

> > > ### Author Response · Authors · 2023-08-13
> > >
> > > We thank the reviewer for responding to our rebuttal. The true (population-level) $\Sigma$ matrix is generated as a random positive definite matrix by drawing a matrix $A \in \mathbb{R}^{n \times p}$ with $N(0,1)$ entries and computing $\Sigma = A^TA + I_p$. We add the identity matrix to ensure that $\Sigma$ is positive definite. We then use the matrix $\Sigma$ to generate the rows of $X$ and $\tilde{X}$ as independent $MVN(0, \Sigma)$ random variables. The use of $\Sigma$ leading to poorer estimation and prediction results (compared to using the matrix $\hat{\Sigma}$ which is directly obtained from discovery data $X$, i.e. the \rho=1 case) is predicted by our theory (see Theorems 3.1 and 3.2). When $\Sigma$ is used as the input to our estimators, this corresponds to the infinitely large proxy data regime with no overlap between the discovery data and proxy data (in other words, $\tilde{n} \rightarrow \infty$ and $\rho = 0$). The form of $\gamma$ in Theorems 3.1 and 3.2 implies that the convergence rate of the estimator in this regime is strictly worse than the optimal minimax rate, which is only achieved if the proxy data and the discovery data are the same. The results of our updated simulation study (see Figure 1 in the newly attached pdf file) support our theory.
> > >
> > > Thank you again for your comment and please let us know if we can provide any further clarification.

---

### Official Review · Reviewer_PbVj · 2023-07-10

**Soundness:** 3 good
**Presentation:** 2 fair
**Contribution:** 2 fair
**Rating:** 6
**Confidence:** 2

**Summary:**

The paper addresses the problem of multi-task learning in settings where only summary statistics (instead of individual-level data) is available, which is a common scenario e.g. in medicine. The paper presents a framework for linear relationships between the covariates and the outcomes which uses summary-level information from distinct sources with a general data-driven tuning scheme for selecting tuning parameters. The results from the theoretical analysis of this framework are confirmed with numerical experiments on synthetically generated data.

**Strengths:**

The paper addresses an important problem for fields where individual level data is not publicly available, which is often the case in health care applications. The numerical experiments confirm the theoretical analysis.

**Weaknesses:**

While the paper addresses an important topic, the presented framework comes with strong limitations. Particularly, the framework can only be applied for linear models which drastically reduces the potential applications of this approach. Additionally, the experimnts are limited to rather simple, synthetic examples that indeed show of the expected behaviour based on the theoretical results, but do not provide any insights on the applicability of the framework in real-world scenarios. Finally, the presentation could be improved (see comments below).

Comments on presentation:
- on related work: first, related work should also be a numbered section (not an unnumbered subsection of introduction). second, there was also quite recently some work by Meija et al. (2022) on estimating causal effects only based on summary statistics reported in medical studies, that seems quite related to the presented problem setup. There the authors show that merging datasets with maximum entropy improves the predictive power compared to using the observed marginal distributions as predictors.
- Line 99: public available -> publicly available.
- After equation (1) you already start using variables with and without tilde ($\tilde{\bf X}$ and ${\bf X}$, $\tilde{\mathcal{D}}$ and $\mathcal{D}$), but you have not really introduced what the difference between them is and only introduce proxy and discovery data (without explaining the terms) in section 3. You should introduce the proxy data (and what that means) much earlier.
- Line 124: in a data-driven in the ... -> in a data-driven way in the ...
- Line 135 and equation above: In the equation use \left( and \right) for the brackets to scale them appropriately. Further, you use $\varepsilon$ as random noise before (above line 88), and now define $\Varepsilon$ as cost of using proxy data that is different from the discovery data. That's confusing. Similarly confusing is that you have now $\tilde{\Sigma}$, $\Sigma_1$ and $\Sigma_2$. Isn't $\Sigma_2$ the same as $\tilde{\Sigma}$? (if not, please clarify what's the difference).
- Line 175: what is $\tilde{p}$? wasn't there only $p$ and $\tilde{\rho}$ before?
- position parameters of figure 2 and 3 should be set to [t]
- Overall structure could be improved. Suggest to have a section on notation and assumptions used throughout the paper before the current section 2. Then you can introduce the problem setup. In that section you right now already mention how summary statistics get included. One could consider doing that only in the next section that focusses on your approach. This way it should be (hopefully) clearer what you propose and what is new about it. Right now it's difficult to understand what are common assumption, what is common knowledge, and what was added on top by you.

**Questions:**

- The framework is designed for multi-task learning with linear relationships between the covariates and the outcomes. Can you provide real-world examples that satisfy this constraint?
- You note that ${\bf X}^{(q)}$ and $\tilde{\bf X}^{(q)}$ are not necessarily the same. Can you give a practical example where they are the same, and one where they are not?
- To obtain equation (2), you replace ${\bf X}^T{\bf X}$ with $\tilde{\bf \Sigma}$, although you before mention that $\tilde{\bf \Sigma}\propto \tilde{\bf X}^T \tilde{\bf X}$, and ${\bf X}$ and $\tilde{\bf X}$ are not the same. Why can you do this replacement here then?

**Limitations:**

authors adequately addressed the limitations and potential negative societal impact of their work

---

> ### Author Rebuttal · Authors · 2023-08-10
>
> We thank the reviewer for excellent feedback about the presentation of our paper, and for raising several important questions. Here we address each of the comments on the paper’s presentation:
> * (Regarding the related work) Thank you for your feedback on formatting and for pointing out the recent work of Meija et al (2022). We will include this paper in our literature in the final version of our paper.
> * (Regarding the comment on Line 99) Thank you for pointing out this typo, this will be fixed.
> * (Regarding the comment on equation (1)) Thank you for this feedback, we will introduce a notation section in the final version of our paper to better clarify the difference between the discovery data X and the proxy data \tilde{X}.
> * (Regarding the comment on Line 124) We will fix this typo.
> * (Regarding the comment on Line 135) We will change the notation from $\mathcal{E}$ to a different character in the accepted version of our manuscript to prevent any confusion with the error term $\varepsilon$. $\Sigma_1$ is the population-level covariance of $X$, meaning that $\Sigma_1 = E[\frac1n X^TX]$. Similarly $\Sigma_2 = E[\frac1n\tilde{X}^T\tilde{X}]$ is the population-level covariance of the proxy data. We will clarify this in the notation section in our final version of this paper.
> * (Regarding the comment on Line 175) Thank you for pointing out this typo, this character should be $\tilde{\rho}$.
>
> **Q1** In our target application, which is genetic risk prediction, genetic variants contribute to observed traits additively, making linear modeling the most effective choice [3,4,5]. We will include a discussion of this assumption in the final version of our paper.
>
> **Q2** In biomedical applications, $X^TY$ can be obtained from studies that focus on marginal
> associations between covariates and the outcome of interest. However, it is often the case that
> these same studies may not report the correlations between the covariates. In this case, such
> information has to be derived from other studies or reference datasets.
> For instance, in genetic studies, many research papers provide GWAS summary statistics
> (https://www.ebi.ac.uk/gwas/), which offer insights into the associations of genetic variants with
> the outcome. However, the correlations between these genetic variants are often reported in
> only a few studies, such as the UK Biobank [2] and the 1000 Genomes data [1].
>
> **Q3** The reviewer is correct that $X$ and $\tilde{X}$ are not the same, and this is due to
> the availability of summary statistics---studies that report $X^TY$ may not report $X^TX$,
> and we need to find a proxy for $X^TX$ from publicly available data. This replacement is how our estimator is defined.
> The main contribution of our theory essentially is to quantify the
> fundamental cost of performing this replacement. If both $X^TY$ and $X^TX$ are
> available from the same study, our theory suggests using the summary stats
> from the same study, instead of using a proxy dataset.
>
>
> [1] The 1000 Genomes Project Consortium et al., “A global reference for human genetic variation,” Nature, vol. 526, no. 7571, pp. 68–74, Oct. 2015, doi: 10.1038/nature15393.
> [2] C. Sudlow et al., “UK Biobank: An Open Access Resource for Identifying the Causes of a Wide Range of Complex Diseases of Middle and Old Age,” PLoS Med, vol. 12, no. 3, p. e1001779, Mar. 2015, doi: 10.1371/journal.pmed.1001779.
> [3] Chatterjee, Nilanjan, Jianxin Shi, and Montserrat García-Closas. "Developing and evaluating polygenic risk prediction models for stratified disease prevention." Nature Reviews Genetics 17.7 (2016): 392-406.
> [4] Torkamani, Ali, Nathan E. Wineinger, and Eric J. Topol. "The personal and clinical utility of polygenic risk scores." Nature Reviews Genetics 19.9 (2018): 581-590.
> [5] Choi, Shing Wan, Timothy Shin-Heng Mak, and Paul F. O’Reilly. "Tutorial: a guide to performing polygenic risk score analyses." Nature protocols 15.9 (2020): 2759-2772.

---

### Official Review · Reviewer_LHqp · 2023-07-27

**Soundness:** 3 good
**Presentation:** 3 good
**Contribution:** 3 good
**Rating:** 5
**Confidence:** 3

**Summary:**

Multi-task learning is a powerful machine learning paradigm for integrating data from multiple sources to improve overall model performance. However, data-sharing constraints in healthcare settings hinder its application. To address this challenge, a flexible multi-task learning framework utilizing summary statistics from various sources is proposed, along with an adaptive parameter selection approach based on a variant of Lepski's method. A systematic non-asymptotic analysis characterizes the performance of the proposed methods under various regimes of sample complexity and overlap. Extensive simulations demonstrate the theoretical findings and the performance of the method, offering a flexible tool for training related models across various domains with practical implications in genetic risk prediction and other fields.

**Strengths:**

Multi-task learning is a promising approach to integrating data from multiple sources and improving individual task performance.
Data-sharing constraints in healthcare and biomedical research limit access to individual-level data, making summary statistics a valuable substitute. The proposed multi-task learning framework allows for simultaneous learning of multiple models using only publicly available summary statistics. A systematic non-asymptotic analysis characterizes the performance of the proposed methods under various regimes of sample complexity and overlap. An adaptive scheme for tuning parameter selection based on a variant of Lepski's method is proposed, allowing for data-driven tuning when only summary statistics are available. The framework has practical applications in genetic risk prediction and can be used to develop trans-ethnic prediction models. The ability to learn from summary statistics offers a versatile tool for developing models across various domains.


Indeed, a major advantage of this method is that it has theoretical guarantees. Through systematic non-asymptotic analysis, this method provides theoretical guarantees for the performance of the multi-task learning framework based on summary statistics, especially under different regimes of sample complexity and overlap. These theoretical guarantees help us better understand the performance of this method and provide guidance for practical applications. Additionally, this method proposes an adaptive parameter selection approach based on a variant of Lepski's method, which can perform data-driven tuning of parameters when only summary statistics are available. The effectiveness of this method is also theoretically guaranteed, which can help us better select parameters and improve the performance of the model in practical applications. Therefore, this method has great practical value and provides a feasible approach for multi-task learning under data sharing constraints.




**Weaknesses:**

1. One major drawback of this method is the lack of comparison with classical multi-task learning methods. While this method provides a novel and useful approach for dealing with data sharing constraints, comparing its performance with existing multi-task learning methods would help to further evaluate its effectiveness. By comparing their performance, we can gain a better understanding of the strengths and weaknesses of these methods and provide better guidance for practical applications. Therefore, future research could consider comparing this method with existing multi-task learning methods.
2. Another potential drawback of this method is that it may not be able to integrate with existing deep learning models. This method is based on training multiple models using basic summary statistics, while deep learning models typically require large amounts of raw data and complex network structures for training. Therefore, integrating this method with existing deep learning models may face many challenges, such as how to convert summary statistics into input data and how to design network structures suitable for summary statistics. While this method is useful in utilizing summary statistics, it may not be applicable in tasks that require processing large amounts of raw data and performing complex computations. Therefore, in practical applications, appropriate methods and models need to be selected based on task requirements.

**Questions:**

1. How does the performance of this method compare to existing classical multi-task learning methods?
2. Can this method be integrated with existing deep learning models, and if so, how?
3. In what type of tasks is this method most suitable, and what type of tasks may require other methods and models?

---

> ### Author Rebuttal · Authors · 2023-08-10
>
> **Q1** Our methods build upon classical multi-task learning techniques, and enable fitting models only using basic summary statistics which are often made publicly available. The sparse $\ell_{2,1}$ regularized estimator extends the group-sparse estimators studied in [4,6], while the nuclear norm estimator expands on the low-rank regression model described in [1]. The nuclear norm approach is closely related to the linear representation learning problem [2,5], constraining regression coefficients to a shared low-dimensional subspace.
>
> The main motivations behind employing these multi-task learning methods but not others are
> threefold: (1) In genetic risk prediction modeling, the additive nature of genetic effects makes
> linear modeling the most effective choice. (2) Across populations and related phenotypes,
> similarities in genetic architectures can be characterized by distance measures in model
> parameters, leading to the enforcement of shared structures through penalty terms. (3) The
> widespread availability of GWAS summary statistics, capturing the marginal correlation between each SNP-phenotype pair, further supports our approach.
>
> In terms of other multi-task learning methods, our general approach of using proxy variables to compute the second-order structure of the predictor variables may be applied to Multi-Task Kernel Ridge Regression in Reproducing Kernel Hilbert Spaces, following [3]. In particular, we may use the proxy variables to compute the kernel matrix between the features. However, the limited availability of publicly accessible proxy kernel matrices in scientific applications hinders the potential usefulness of this method.
>
>
>
> **Q2** In real-world applications, fitting a deep learning model with only $X^TX$ and $X^TY$ available
> poses significant challenges, if not impossibilities. The summary stats represent highly
> condensed summaries of linear relationships, whereas the effectiveness of deep learning
> models lie in their capacity to capture non-linear relationships. Hence, relying solely on such
> information is intuitively insufficient for successful deep learning model training.
>
> Nevertheless, we hold the anticipation that training neural models could be feasible if new classes of summary statistics are created and incorporated. This approach would require practitioners to explore and develop innovative summary-level information.
>
> In a more extreme scenario, the problem could be formulated as a federated learning problem [7], wherein data owners iteratively share gradient-type information for model updating. However, this approach differs from our motivation, which aims to leverage pre-existing summary stats instead of requiring continuous sharing during model training.
>
> An intriguing avenue worth exploring is the integration of external basic summary statistics to assist in the training of neural models using internal datasets. This direction is currently under investigation, and we are actively exploring its potential implications.
>
>
> **Q3** Our method is particularly well-suited for applications where the dominant signals exhibit
> linearity. For instance, in genetic risk prediction modeling, linear models have proven to be
> effective. This choice is driven by the constraint of utilizing only publicly available summary
> statistics. We advocate that in many scenarios, linear models serve as robust and stable
> working models. With our method, researchers without access to individual-level data sources
> can still gain insight from summary statistics in terms of classification or risk modeling.
> Moving forward, we recognize the importance of developing novel methods and theories for
> other canonical statistical problems. These endeavors may necessitate summary statistics in
> new forms, tailored to specific tasks, and represent crucial areas for future research.
>
> [1] S. Negahban and M. J. Wainwright, “Estimation of (near) low-rank matrices with noise and high-dimensional scaling,” Ann. Statist., vol. 39, no. 2, Apr. 2011, doi: 10.1214/10-AOS850.
> [2] S. S. Du, W. Hu, S. M. Kakade, J. D. Lee, and Q. Lei, “Few-Shot Learning via Learning the Representation, Provably.” arXiv, Mar. 30, 2021. Accessed: Feb. 10, 2023. [Online]. Available: http://arxiv.org/abs/2002.09434
> [3] Charles Micchelli and Massimiliano Pontil, “Kernels for Multi-task Learning,” Advances in Neural Information Processing Systems, vol. 17, 2004.
> [4] K. Lounici, M. Pontil, S. van de Geer, and A. B. Tsybakov, “Oracle inequalities and optimal inference under group sparsity,” The Annals of Statistics, vol. 39, no. 4, pp. 2164–2204, Aug. 2011, doi: 10.1214/11-AOS896.
> [5] N. Tripuraneni, C. Jin, and M. I. Jordan, “Provable Meta-Learning of Linear Representations.” arXiv, Dec. 31, 2021. Accessed: Sep. 08, 2022. [Online]. Available: http://arxiv.org/abs/2002.11684
> [6] K. Lounici, M. Pontil, A. B. Tsybakov, and S. van de Geer, “Taking Advantage of Sparsity in Multi-Task Learning.” arXiv, Mar. 08, 2009. Accessed: Sep. 08, 2022. [Online]. Available: http://arxiv.org/abs/0903.1468
> [7] M. I. Jordan, J. D. Lee, and Y. Yang, “Communication-Efficient Distributed Statistical Inference,” Journal of the American Statistical Association, vol. 114, no. 526, pp. 668–681, Apr. 2019, doi: 10.1080/01621459.2018.1429274.

---

### Author Rebuttal · Authors · 2023-08-10

We would like to thank the reviewers for their valuable and insightful feedback. In this global response we would like to describe the major changes we made to address the reviewers’ comments, which greatly improved the quality of our work.

**First, we add a new simulation study evaluating our method based on out-sample prediction accuracy.** The setup of the simulation is the same as what has been considered in the previous submission. We simulated a dataset with $n_{min} = 100$, $p = 100$,  $\tilde{n} = \tau n_{min}$ for $\tau \in (0.5, 1, 5, 10)$ and $\tilde{\rho}_q= 0$ for each q. The number of tasks was fixed at 8. We generated a row-sparse matrix with 10 nonzero rows and a rank 2 $B^*$ for the sparse and low-rank multi-task estimators. After fitting our model and obtained the parameter estimate for $B^*$ , we generate a new test set of size 100 according to the same data generating process and evaluate our estimator's ability to predict the outcome in this test set. The results are given in the first figure in our new additional PDF document. We found that the relative performance of our method compared to the individual level estimator and the true covariance estimator is similar to the estimation task we showed in our previous submission. The performance of the proxy data estimator improves as the size of the proxy data grows, matching that of the oracle estimator that uses the true covariance matrix as its input. We observe the same performance gap between the proxy data estimator and the individual-level data estimator.

**Secondly, we have applied our method to analyze real genetic data to demonstrate the real-world applicability of our method.** We use a multi-site data obtained from the electronic Medical Records and Genomics (eMERGE) network [1], which includes individual-level genotype data from multiple research sites in the United States. Our goal is to predict levels of low-density lipoprotein (LDL) across five adult sites, treating the data from each site as a separate task. We split the data (with sample sizes $n_1 = 3813, n_2 =  546, n_3 = 2666, n_4 = 1435, n_5 = 525$) at each task into a training and test set (with a test set data size of 100 for each task) and evaluate the performance of our method using the prediction MSE on the test set.
The training data from each site is used to construct the discovery summary statistics $X_q^TY_q$ for each task. For approximating $\Sigma_q$, we choose two different approaches: one is to use the half of the genotype data from each site (this approach is labeled as Proxy_MTL1); the other approach is to use $X_1$ (genotype data from site 1) to approximate $\Sigma_q$ for all the sites. This approach is labeled as Proxy_MTL2. We use these two approaches to demonstrate a potential trade-off in the construction of the reference panel: Proxy_MTL1 uses a well-specified reference dataset with a smaller sample size; Proxy_MTL2 uses a larger reference dataset that may suffer from a distribution shift. For comparison, we also fit a multi-task learning estimator that uses all of the individual level training data for each task, which is labeled ‘Individual_MTL’, and we fit a ridge regression estimator that models each task separately, for comparison to our multi-task learning approach. The ridge estimator uses the proxy sample covariance instead of the individual level covariance matrix for a fair comparison with our method. We repeat the train-test split process 10 times, which admits the distribution of prediction MSE values that we report in the figure. We use the nuclear norm penalized multi-task learning estimators in this application, because we believe that the genetic effects in this dataset are dense.

Our results demonstrate that our method is highly practical when only summary-level information is available, as the prediction MSE of our method is nearly the same as the estimator which uses the individual-level data, despite a slight cost in performance. Furthermore, all multi-task learning estimators outperform the ridge-estimator, confirming that multi-task learning is a strong approach when there is shared structure between tasks.

**Finally, we clarified the motivation and provided real-world justifications for considering the discovery data to be different from the proxy data, and we clarified our technical innovation.** Due to the space limit, we refer it to the point-to-point responses to each reviewer.

Once again, we thank the reviewers for their comments, and we look forward to further discussion.

[1] McCarty, C.A., Chisholm, R.L., Chute, C.G. et al. The eMERGE Network: A consortium of biorepositories linked to electronic medical records data for conducting genomic studies. BMC Med Genomics 4, 13 (2011). https://doi.org/10.1186/1755-8794-4-13

---

### Decision · Program_Chairs · 2023-09-21

**Decision:**

Accept (poster)

**Comment:**

This paper addresses the problem of multi-task learning where only summary statistics available. This setting arises in medical or biological applications such as genetic risk prediction. The approach is limited to a linear setting. A theoretical analysis and an evaluation are provided.


The reviewers have acknowledged that the paper studies an important an interesting problem and the theoretical results are a strong point of the paper. Some reservations were also raised on the limited experimental evaluation, on the limitation to linear models, and on the paper presentation.


During rebuttal, authors have addressed the points raised by the reviewers, including:
(i) a new experimental study (simulation study on out-sample accuracy, analysis of real genetic data).
(ii) some modifications to improve the presentation of the paper.
(iii) the addition of a little discussion about the lack of comparison with other multi-task learning methods, the interest of the method for applications where dominant signal exhibits linearly (reviewer LHqp).
(iv) a justification of the linear modeling as the most effective choice for the target task genetic risk prediction, as well as with a comment on  the potential use of GWAS summary statistics in biomedical applications (reviewer PbVj).
(v) an answer on the difference between $X$ and $\var{X}$ (reviewers PbVj and Z4Yp).
(vi) a discussion on the perspective of integration with deep learning models (reviewers LHqp and Z4Yp).
(vii) an explanation to the behavior in Figure 1 commented by reviewer Z4Yp.

Overall, authors have answered mostly to the issues raised by reviewers. The global evaluation is positive.
I propose then acceptance.
I encourage the authors to add all the elements provided in the answers of the rebuttal in the final version of the paper